# Exploring the values and preferences of children and adolescents with obesity and their parents/caregivers concerning diet or physical activity interventions for weight management: Mega-ethnography of qualitative syntheses

**Munira Essat** [1]*, **Christopher Carroll**[1], **Andrew Booth**[1], **Joanna Leaviss**[1], **Diana Castelblanco Cuevas**[1], **Roos Verstraeten**[2]

**1** Sheffield Centre for Health and Related Research, School of Medicine and Population Health, University of Sheffield, Sheffield, United Kingdom, **2** European Commission – Joint Research Centre, Ispra, Italy

* m.essat@sheffield.ac.uk

## Abstract

### Introduction

Obesity and overweight are a global health problem affecting over 2.6 billion people worldwide. Diet and physical activity, including structured exercise, are critical components in the management of obesity. This review of reviews explores the values and preferences of children, adolescents, and their parents/caregivers that influence engagement and adherence to dietary and physical activity interventions for obesity management.

### Methods

Eleven electronic databases were searched (from January 2010 to June 2024) to identify reviews incorporating qualitative research on values and preferences, attitudes or experiences of children or adolescents with obesity, or their caregivers, in relation to diet or physical activity interventions. Supplementary grey literature searching, citation searching and reference lists screening of included reviews were also undertaken. The methodological quality of included reviews was appraised using the Swedish Agency for Health Technology Assessment (SBU) tool. Data synthesis was performed using a mega-ethnography approach.

### Results

Fourteen reviews were included with majority of studies conducted in high-income countries. Six key factors were identified that affect children and adolescents or their caregivers when initiating and continuing with physical activity and/or dietary management interventions. These included perceptions concerning the value of

**Data availability statement:** All relevant data are within the paper and its Supporting Information files.

**Funding:** The Department of Nutrition and Food Safety at the World Health Organization (WHO) commissioned and provided financial support to the University of Sheffield for this work. WHO acknowledges the financial support from the Norwegian Agency for Development Cooperation (NORAD), The Swedish International Development Cooperation Agency (SIDA), The Government of the Grand Duchy of Luxembourg, and the Government of Germany (BMG) to the Department of Nutrition and Food Safety. The funders had no role in study design, data collection and analysis, decision to publish, or preparation of the manuscript. There was no additional external funding received for this study.

**Competing interests:** The authors have declared that no competing interests exist.

interventions; competing priorities; the role of social support; the physical environment; the nature and content of the intervention; and costs.

## Conclusion

Multiple factors influence engagement with diet and physical activity programmes among children and adolescents with obesity and their caregivers, highlighting the need for emotional and psychological support, whole-family involvement, and personalised, trust-based guidance from health professionals. However, these findings are context-dependent with evidence primarily from high-income countries, which may limit their wider generalisability.

## Introduction

In 2020, it was estimated that 2.6 billion people globally were overweight and obese (as classified by BMI ≥ 25 kg/m$^2$) [1,2]. Recent projections suggest this number will increase to over 4 billion, totally over 50% of the global population, by 2035 [1,2]. The prevalence of overweight and obesity among children and adolescents aged 5–19 years has risen dramatically, from 4% in 1975 to just over 18% (340 million) in 2016 [3]. Projections indicate a further increase globally of 10% to 20% among boys and 8% to 18% among girls by 2035 [1,2]. Similarly, rates of overweight and obesity are also rising in children under the age of 5 years, with an estimated 39 million in 2020 [4]. Obesity contributes to multiple comorbid and diet-related chronic medical conditions, including type 2 diabetes and some forms of cancer [5]. A failure of prevention, management or treatment interventions could lead to a global economic impact of $4.32 trillion per year by 2035, with nearly 3% of global Gross Domestic Product (GDP) treating illnesses related to obesity [2]. No country is currently on track to halt the rise of obesity and meet the World Health Organization (WHO) target of 'no increase' on 2010 levels by 2025 [1,2]. Likewise, very few countries are on course to meet the global targets for diet-related non-communicable diseases, including obesity [1,2,6].

To strengthen efforts in preventing, managing and treating obesity across the life course, WHO recognized the need to supplement existing prevention-focused guidelines with normative guidance focusing on people-centred integrated management and care of children and adolescents with obesity. This approach emphasizes primary health care services that provide health and obesity management advice and guidance to children with obesity and their families. Understanding the values and preferences of patients and their families is central to this people-centred approach. Research from community-engaged models and translational science consistently shows that interventions are more likely to be adopted, effective, and sustainable when they align with the lived experiences, needs, and priorities of the communities they are intended to serve [7,8]. To support these efforts, the WHO commissioned a series of parallel reviews to underpin guidance on this issue. These reviews examined global evidence from children and adolescents with obesity, as well as their

parents and caregivers, concerning diet, physical activity (including exercise), behavioural, pharmacological and bariatric surgery or weight loss device interventions for managing obesity. This paper reports the findings of the reviews that examined the evidence related to (i) dietary interventions for children and adolescents living with obesity, and (ii) physical activity interventions for children and adolescents living with obesity. Although various reviews have synthesized the evidence on obesity management strategies involving either diet and/or physical activity, none have focused exclusively on values and preferences of children, adolescents, and/or their parents/caregivers, that influence engagement and adherence to dietary and physical activity interventions for obesity management in children and adolescents [9]. Yet, such perspectives are essential for understanding why children and families choose to engage or not engage with weight management interventions, and what helps or hinders continued participation. Therefore, to provide evidence-based guidance for WHO and to inform the development of future interventions, we report the first review of systematic reviews exploring the values and preferences of children and their parents/caregivers globally regarding (i) dietary interventions and (ii) physical activity interventions. Specifically, this review was guided by the following overarching questions: (i) What values and preferences do children, adolescents, and their parents/caregivers express regarding dietary and physical activity interventions for obesity management? (ii) How do these values and preferences influence engagement, adherence, and dropout? These questions informed the search, eligibility criteria, and synthesis approach.

## Methods

This review of reviews used the EPOC's Protocol and Review Template for Qualitative Evidence Synthesis to ensure a consistent and coherent approach [10]. Pre-specified review protocols were registered with the PROSPERO international prospective register of systematic reviews [diet: CRD42022312881 and physical activity: CRD42022310922]. This review did not require separate ethical approval, as it involved secondary analysis of anonymised data from previously published studies. Accordingly, all included primary studies had obtained appropriate ethical approval.

### Eligibility criteria

To be included in this combined intervention review (review of physical activity interventions and review of dietary interventions), reviews had to satisfy the topic of interest, defined here using the PerSPECTiF framework [11], see Table 1.

For inclusion, reviews had to include at least one primary qualitative study and collect and analyse qualitative data using qualitative data collection and analysis methods. Reviews that used mixed methods approach were included where it was

**Table 1. Problem definition using PerSPECTiF framework [11].**

| Perspective | Setting (Intervention delivery) | Phenomenon of Interest | Environment | Comparison | Time/ Timing | Findings |
|---|---|---|---|---|---|---|
| Children with obesity | Home School Clinical settings | Diet Intervention; Physical Activity Interventions | Global (High, Middle- and Low-Income Countries) | PROGRESS+ Factors | Initiation Continuation | Themes Values Preferences Attitudes |
| Adolescents[a] with obesity | | | | | | |
| Parents and Caregivers of children and adolescents with obesity? | | | | | | |

[a]The term "adolescent" is defined by the World Health Organization (WHO) as a person aged 10–19 years old, and "youth" as one between the age of 15–24 years old. [12]

Abbreviations: PROGRESS+ Factors, Place of residence; Race/ethnicity/ culture/ language; Occupation; Gender/sex; Religion; Education; Socioeconomic status; Social capital

+ refers to:

1) personal characteristics associated with discrimination (e.g., age, disability)

2) features of relationships (e.g., parents who smoke, exclusion from school)

3) time-dependent relationships (e.g., leaving the hospital, respite care, other instances where a person may be temporarily at a disadvantage)

possible to extract qualitative data from studies that had been collected and analysed using qualitative methods. Reviews could be either published or unpublished and not limited by language. We did not apply any 'quality threshold' for inclusion.

Physical activity interventions specifically relate to provision of selected and/or *structured* physical activity options for individuals with obesity, either delivered in an individual or group setting. Dietary interventions specifically relate to any intervention for dietary management with the aim of managing obesity, including nutritional and diet options, which could be delivered in individual or group format. Studies from January 2010 onwards were included to ensure the most contemporaneous evidence (although reviews published during this period would inevitably capture earlier relevant primary studies).

The focus of this review was on qualitative data describing the values and preferences of children and adolescents, and of their parents/caregivers, regarding these types of interventions generally. Reviews focusing only on a specific type of exercise or physical activity, or composition of diet were excluded. Furthermore, reviews exploring diet or physical activity for prevention of obesity (i.e., in population who were not obese) were excluded. The views of other adults involved in dietary management or physical activities, e.g., clinicians, schoolteachers or other staff, although acknowledged as important, were excluded from the scope of the review.

## Search methods for identification of studies

The search strategies for each database were based on previous reviews conducted on the topic of obesity [13]. The search approach is presented in Table 2 and was documented using the STARLITE elements (Sampling strategy, Type of study, Approaches, Range of years, Limits, Inclusions and Exclusions, Terms used, Electronic sources). Searches for reviews were conducted in January 2022 and updated in June 2024. A methodological filter was used to populate the Cochrane Qualitative and Implementation Methods Group register of qualitative evidence syntheses. Additional

**Table 2. Search approach using the STARLITE elements.**

| | What matters to children and adolescents with obesity and their parents and caregivers concerning diet or physical activity interventions for children and adolescents with obesity? |
|---|---|
| **Sampling strategy** | Comprehensive: samples from international and regional databases |
| **Type of study** | Systematic reviews that include qualitative research studies |
| **Approaches** | Backwards citation chasing (reference checking) and Forwards citation chasing (Google Scholar citation searching using Publish or Perish) |
| **Range of years** | January 1, 2010 – June 2024 |
| **Limits** | No language limits; Human only |
| **Inclusions and Exclusions** | Inclusion (Mega-aggregation): Reviews reporting perceptions of children, parents or caregivers concerning intervention for dietary management or physical activity; desires, fears, experiences, coping strategies etc., Reviews exploring initiation, contemplation or continuation of diet or physical activity interventions or services |
| **Terms used** | See Appendix 1 for full strategy. Conceptualised as Obesity concepts AND Child/Adolescent/Parent terms AND Qualitative Review labels |
| **Electronic sources** | 11 data sources: AJOL; ASSIA; CINAHL (Ovid); EMBASE (Ovid); EPISTEMONIKOS; Google Scholar; LILACS; MEDLINE (Ovid); PsycINFO (Ovid); Scopus; WoS |

Abbreviation: AJOL, African Journals Online; ASSIA, Applied Social Sciences Index and Abstracts; CINAHL (Ovid), Cumulative Index of Nursing and Allied Health Literature; LILACS, Latin American and Caribbean Health Sciences Literature; WoS, Web of Science.

supplementary searches included a grey literature search in the following sources: OpenGrey, Grey Literature Report, Agency for Healthcare Research and Quality, National Institute for Health and Clinical Excellence website and Eldis; citation reference search for all included studies; checking reference lists of reviews from a previous mega-aggregation conducted for the WHO [13]; and of all the included reviews and key references. We also actively pursued related studies identified through shared authorship, citation networks or related articles features [14]. The Ovid MEDLINE search strategy, which was adapted for other databases is presented in S1 Appendix. The same search strategy was used to identify reviews for physical activity interventions and reviews for dietary management interventions.

## Selection of studies

Once an agreement was reached on the eligibility criteria during pilot screening of 100 references, the review team proceeded with the title and abstract screening. The remaining references were equally divided among the reviewers. To ensure consistency, a second reviewer checked 10% of the references excluded by each primary reviewer. Any discrepancies were discussed and resolved collaboratively by the entire review team. Full text articles was obtained for all references deemed potentially relevant. These articles were independently assessed for final inclusion by the lead reviewer and an additional, independent reviewer. Disagreements were resolved through discussion or, when required, by involving a third reviewer to reach consensus.

## Data extraction

Data was extracted in a pre-piloted data extraction form by one reviewer and checked by a second reviewer (CC or ME). Inconsistencies were resolved by discussion and, if necessary, consultation with a third reviewer (AB). The following data were extracted: author; year; aim of review; type of review; databases searched; inclusion criteria (age groups and regions/countries covered); number of included qualitative and mixed-method studies; appraisal tool used; type of synthesis conducted; potential additional relevant references. The data extraction form also captured synthesis-related data, including: (1) the overarching themes identified in the review (third-order constructs – The reviewers' synthesis or interpretation of the second-order constructs across studies), (2) themes reported by the authors of the primary studies (second order constructs – The interpretations or themes developed by the authors of the original studies); and (3) supporting participant quotes or data from the primary studies (first order constructs).

## Quality appraisal

The methodological limitations of the included reviews of qualitative evidence was assessed using the Swedish Agency for Health Technology Assessment (SBU) tool [15] by one reviewer and checked by a second reviewer. Disagreements were resolved by discussion. Data richness is an important consideration within qualitative research. We applied a previously used Qualitative Evidence Syntheses (QES) rating scale [16], modified from a scale originally developed for primary qualitative studies [17]. This 3-point score system considers the amount and depth of qualitative data and its relevance to the research question. A QES is scored 3 (the highest score of data richness) when it includes large quantities of qualitative studies (>20) or qualitative data (that is, illustrative quotations from primary supporting studies); 2 for a substantive quantity of qualitative studies (10–20) or qualitative data; and 1 (lowest data richness score) if there are few qualitative studies (<10) or very little or no qualitative data.

## Data analysis and synthesis

Synthesis was performed using a mega-ethnography approach [18]. Mega-ethnography requires the extraction of diverse findings from included reviews: first order constructs (any relevant participant verbatim comments from the original primary research studies); second order constructs (primary research authors' statements of findings); and third order constructs

(the review authors' own themes of statements of findings emerging from their synthesis). Constructs were extracted independently by a single reviewer and checked by the second reviewer (CC or ME), inconsistencies were resolved by discussion and, if necessary, consultation with a third reviewer (AB). In this process, all relevant themes or data that related to diet and/or exercise reported in each included review were transferred verbatim into an Excel spreadsheet and labelled as first, second or third order constructs. Synthesis then involved the interpretation and categorization of the third order constructs by one reviewer (CC or ME), using details provided in any first and second-order constructs also extracted, to develop and group them under new, fourth-order constructs. These new constructs were then grouped into higher-level themes to develop the review findings. These new themes and their supporting data were critically reviewed by the review team and revisions made if necessary.

### Review author reflexivity

As a review team, we were very aware of the need to consider our own biases when conducting this work. A full reflexivity statement is available in S2 Appendix.

## Results

### Results of the search

A total of 3343 titles and abstracts were screened, and 146 papers were selected for full-paper screening. From these, fourteen reviews and syntheses were identified that satisfied the inclusion criteria [19–32]. Of these, seven reviews [21,23–27,29] were included in both the dietary management review and physical activity review; four reviews [19,22,31,32] were included in the physical activity review only; and three reviews [20,28,30] were included in the dietary management review only. The overall process is detailed in the PRISMA flow diagram (Fig 1). Details of reviews excluded at full text stage, with reasons for exclusion are available in S1 Table.

### Description of the included reviews

Details of the included reviews are presented in Tables 3 and 4. Further details are provided in S2 and S3 Tables. The fourteen included reviews or syntheses [19–32] were published in English between 2013 and 2024. One review [21] included only studies of children (up to the age of 11 years); five reviews [19,22,23,28,32] included only studies of adolescents (aged 9–18 years); one review [20] did not specify the age. The remaining seven reviews [24–27,29–31] included studies relating to both children and adolescents (from 0 to 18 years). Eleven [19,21,23–30,32] of the fourteen reviews included study populations from Europe, eleven [19,20,23–30,32] from North America, and eight [19,22,23,25–28,30] from the Western Pacific region (Australia and New Zealand), including one review [22] that drew upon evidence exclusively from Western Pacific region. Only three reviews included primary studies from Africa, [19,26,28] and four from south-east Asia [23,27,28,32]. One review did not report the origins of the included studies [31]. The evidence was predominantly from high-income countries.

The number of primary qualitative studies in total, reported as included, across the reviews was 227, and the number of primary qualitative studies included in any review ranged from two [22] to 48 [28]. The included reviews had searched between three [20] and 11 [21] bibliographic databases. The majority of reviews conducted thematic synthesis.

### Quality and richness assessment of included reviews

Details of the quality and richness assessments are presented in Table 5. The methodological assessment judged six of the reviews to have been conducted with only a minor risk to rigour [19,23–27]; for six reviews the risk to rigour was deemed to be moderate [20–22,28,31,32]; and for two reviews the risk to rigour was assessed as high [29,30]. Three reviews were considered to be supported by rich data [23,26,28], two of which were also assessed as being conducted

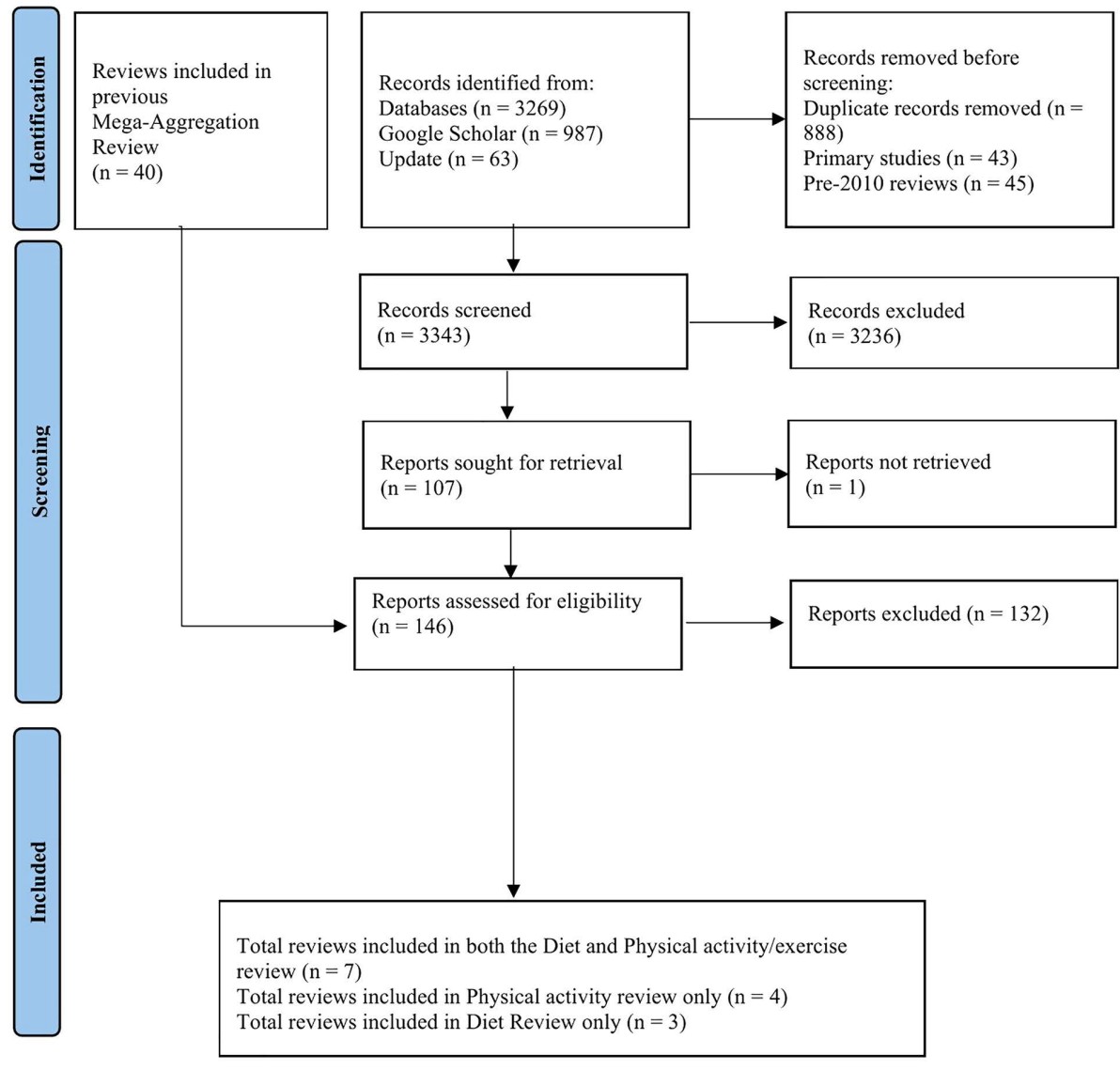

**Fig 1. PRISMA 2020 flow diagram for eligible article identification [33].**

with only a minor risk to rigour [23,26] and four were deemed to be supported by moderately rich data [19,21,27,31]. The data were assessed as meagre in the remaining seven reviews [20,22,24,25,29,30,32]. Full details of quality assessment are provided in S4 Table.

## Review findings

A summary of the review findings (the fourth-order constructs) and a list of the supporting reviews for each finding are presented in Table 6. Full details of the findings of the synthesis: illustrative text extracts, the novel, fourth-order constructs, their basis in the third-order constructs of the included reviews, and the details of the reviews underpinning each finding, are presented in S5-S10 Tables. A narrative detailing each of the findings is presented below.

**Table 3. Characteristics of included reviews.**

| First Author (year of publication) | Review type | Age of Children (review inclusion criteria) | Regions | Number of Qualitative studies (Number of total studies in the review) | Method of Synthesis |
|---|---|---|---|---|---|
| **Dietary management and physical activity management** | | | | | |
| **Burchett (2018) [21]** | Systematic review using QCA | 0-11 | Europe | 11 (11) | Thematic analysis |
| **Jones (2019) [23]** | QES | 9-18 | North America, Europe, South East Asia, Western Pacific | 24 (28) | Thematic Synthesis |
| **Kebbe (2017) [24]** | Thematic synthesis | 2-18 | North America, Europe | 11 (17) | Thematic synthesis |
| **Kelleher (2017) [25]** | Systematic review | 2-18 | North America, Europe, Western Pacific | 6 (13) | Narrative synthesis |
| **Lachal (2013) [26]** | QES | 0-18 | Africa, North America, Central America, Eastern Mediterranean, Europe, Western Pacific | 45 (45) | Meta-synthesis, adapted from meta-ethnography |
| **Lang (2021) [27]** | QES | 2-18 | North America, Europe, South East Asia, Western Pacific | 16 (16) | Thematic synthesis |
| **Roberts (2021) [29]** | Other – Scoping review | 2-18 | North America, Europe | 9 (12) | Summative content analysis (SCA) |
| **Physical activity management (including exercise) only** | | | | | |
| **Chen (2024) [31]** | Systematic review | 6-18 | NR | 15 (31) | Thematic synthesis |
| **Haracz (2013) [22]** | Integrative review | 9-18 | Western Pacific | 2 (22) | Thematic synthesis |
| **Skogen (2022) [32]** | Scoping review | 13-18 | North America, Europe, South East Asia | 6 (12) | Thematic synthesis |
| **Stankov (2012) [19]** | QES | 9-18 | Africa, North America, Europe, Western Pacific | 15 (15) | Miles and Huberman's cross-case analysis integrated with thematic network analysis |
| **Dietary management only** | | | | | |
| **Liu (2021) [28]** | QES | 9-18 | Africa, North America, Central America, Eastern Mediterranean, Europe, South East Asia, Western Pacific | 48 (48) | Thematic synthesis |
| **Molina (2021) [20]** | Scoping review | NR | North America | 10 (44) | Narrative synthesis |
| **Zarnowiecki (2020) [30]** | Systematic review | >1 years | North America, Europe, Western Pacific | 9 (35) | Narrative synthesis |

Abbreviations: NR, Not reported; QCA, Qualitative Comparative Analysis; QES, Qualitative evidence synthesis; SCA, Summative content analysis

Fig 2 illustrates the factors that influence whether children and adolescents with obesity, or their caregivers, are willing or able to engage or continue participating over time in physical activity programs or dietary interventions.

Interventions to manage weight loss based on the views and experiences of children and adolescents with obesity and their caregivers.

## Perceptions of the value of interventions

**Children and adolescents with obesity can be motivated to engage with obesity health services that focus on diet or physical activity interventions because of the perceived benefits of such activities.** Children and adolescents with obesity

**Table 4. Details of review methods used in the included reviews.**

| First Author (year of publication) | Search Dates | Total Number of Databases Listed | Main Databases Used | Additional Databases | Supplementary Search (i.e., non-Database) Methods | Quality Assessment Tool(s) | GRADE-CERQual Used |
|---|---|---|---|---|---|---|---|
| **Dietary management and physical activity management** | | | | | | | |
| **Burchett (2018) [21]** | 1990 – March 2016 | 11 | MEDLINE, CINAHL, PsycINFO, ASSIA, Web of Knowledge/ Science Citation Index/ Social Science Citation Index, British Education Index, Education Resources Information Centre (ERIC), Index to Theses in Great Britain and Ireland, HMIC, PubMed, Social Policy and Practice | HMIC, PubMed, Social Policy and Practice | Studies from existing reviews with update search in December 2015 – details not reported | Other | No |
| **Jones, (2019) [23]** | Database inception to July 2018 | 6 | MEDLINE, CINAHL, EMBASE, PsycINFO, ASSIA, Web of Knowledge/ Science Citation Index/ Social Science Citation Index | None | Relevant systematic reviews, key journals and reference lists of included studies were manually screened. A specialist librarian was consulted to refine the search. | EPPI-Centre Tool | No |
| **Kebbe (2017) [24]** | 1980–June 2016 | 6 | CINAHL, EMBASE, MEDLINE, PsycINFO, ProQuest Dissertations and Theses and Scopus) | None | RLIS | MMAT | No |
| **Kelleher, (2017) [25]** | Database inception to 2015 | 4 | MEDLINE, CINAHL, EMBASE, PsycINFO | None | RLIS | Other - Bowling's quality checklist | No |
| **Lachal (2013) [26]** | 1990–2011 | 5 | MEDLINE, CINAHL, EMBASE, PsycINFO, Scopus | None | RLIS | Other | No |
| **Lang (2021) [27]** | Database inception to January 2019; January 2019 to April 2020 | 6 | MEDLINE In-Process and other nonindexed citations, EMBASE, CINAHL Plus, PsycINFO, Ovid Emcare, and Scopus | None | RLIS | CASP | No |
| **Roberts (2021) [29]** | 2006 -February 2018 | 6 | PubMed, CINAHL, Scopus, PsycINFO, Cochrane Reviews, and Embase | None | An ancestral search of reference lists from seminal papers | None | No |
| **Physical activity management (including exercise) only** | | | | | | | |
| **Chen (2024) [31]** | 2003–2023 | 3 | Web of Science, Scopus, and PubMed | None | None | CASP | No |

*(Continued)*

**Table 4.** (Continued)

| First Author (year of publication) | Search Dates | Total Number of Databases Listed | Main Databases Used | Additional Databases | Supplementary Search (i.e., non-Database) Methods | Quality Assessment Tool(s) | GRADE-CERQual Used |
|---|---|---|---|---|---|---|---|
| **Haracz (2013) [22]** | 2002-2012 | 4 | CINAHL, Medline, AMED and PsycINFO, | None | Searches of nine occupational therapy journals | McMaster University Guidelines and Appraisal Forms for Quantitative and Qualitative Research | No |
| **Skogen 2022 [32]** | Inception to 2020 | 4 | PubMed, Web of Science, SportDiscuss, and Cinahl | | Reference list of all included studies and reviews found through database searches | None | No |
| **Stankov (2012) [19]** | 1950–2009 | 6 | MEDLINE, CINAHL, EMBASE, PsycINFO, SportsDiscus and Academic Search Premier | None | Searches of non-peer reviewed 'grey' literature (i.e., Google Scholar, World Wide Web and professional networks) and pearling reference lists | CASP | No |
| **Dietary management only** | | | | | | | |
| **Liu (2021) [28]** | Database inception to 31 July 2020 | 4 | PubMed, Web of Science, PsycINFO, and EMBASE | None | RLIS | Adapted CASP | No |
| **Molina (2021) [20]** | Database inception to February to April 2020. | 3 | PubMed, LILACS and EconLit | Governmental websites of each of the 19 countries (e.g., Ministry of Health), Governmental press releases or other official documents. websites of WHO, PAHO and the World Cancer Research Fund International (WCRFI) Google Scholar | Reference list of all retrieved documents or publications | None | No |
| **Zarnow-iecki (2020) [30]** | 1946 to October 22, 2018 restricted 2013–2018 | 5 | MEDLINE (Ovid capturing PubMed) and translated for use in EMCARE (Ovid), PsychINFO (Ovid), Scopus and Pro-Quest databases. | None | RLIS and relevant reviews identified in the search | None | No |

Abbreviations: CASP, Critical Appraisal Skills Programme; MMAT, Mixed Methods Assessment Tool; NR, Not Reported; RLIS, Reference Lists of Included Studies

Table 5. Quality Assessment (risk to rigour evaluated using the SBU tool) and Richness Assessment.

| | Bur-chett 2018 [21] | Chen 2024 [31] | Jones 2019 [23] | Kebbe 2017 [24] | Kelle-her 2017 [25] | Lachal 2013 [26] | Lang 2021 [27] | Rob-erts 2021 [29] | Haracz 2013 [22] | Stan-kov 2014 [19] | Sko-gen 2022 [32] | Liu 2021 [28] | Molina 2021 [20] | Zarnow-iecki 2020 [30] |
|---|---|---|---|---|---|---|---|---|---|---|---|---|---|---|
| 1. Aim | L | M | L | L | L | L | M | L | M | L | L | L | L | L |
| 2. Search approach | M | L | L | L | L | L | L | L | L | L | L | L | L | L |
| 3. Inclusion criteria | L | L | L | L | L | L | L | L | L | L | L | L | M | L |
| 4. Competence | M | M | M | L | M | M | L | H | L | M | L | M | M | H |
| 5. Search strategy | M | L | L | L | L | L | M | M | M | L | L | M | H | L |
| 6. Study screening | L | L | L | L | M | L | L | H | L | M | L | L | M | L |
| 7. Appraisal tool | H | L | L | M | L | L | L | H | L | L | H | M | H | M |
| 8. Appraisal process | H | L | L | L | L | M | L | H | H | L | H | H | M | L |
| 9. Synthesis (method appropriateness) | L | L | L | L | L | L | H | M | L | L | M | L | L | M |
| 10. Synthesis process | L | L | L | L | M | L | H | M | M | L | M | L | L | M |
| 11. Synthesis output: clearly grounded in primary studies | L | L | L | L | L | L | H | L | H | L | L | L | L | H |
| 12. Synthesis output: beyond a summary of primary studies | H | M | M | L | L | M | L | L | M | L | M | L | H | H |
| 13. Confidence in finding (CERQual) | H | H | L | H | H | H | H | H | H | H | H | H | H | H |
| Overall verdict (concerns) | Mod-erate | Mod-erate | Minor | Minor | Minor | Minor | Minor | High | Mod-erate | Minor | Mod-erate | Mod-erate | Moder-ate | High |
| Data richness score | 2 | 2 | 3 | 1 | 1 | 3 | 2 | 1 | 1 | 2 | 1 | 3 | 1 | 1 |

H: High risk to rigour. M: Moderate risk to rigour. L: Low risk to rigour.

Data richness: 3=rich data; 2=moderately rich data; 1=meagre data.

reported being motivated by several factors to engage with the interventions: not only the expected and hoped-for benefits of weight loss [27,32], but also other benefits, such as preventing health sequelae, being socially accepted, and building self confidence and self-esteem [23,25,27,28,31,32]. Adolescents highlighted that their desire for weight loss was partly driven by wanting to take responsibility/ have agency for their condition and to follow a healthy lifestyle [23]. Some children and adolescents also reported being motivated by observing the achievements of other children and adolescents who have engaged, or are engaging, with such activities [27], as well as recognising their own success [23,31]. Other factors that particularly motivated adolescents to engage with physical activity include being able to attend a gym [23,24]. Benefits differ by gender, with girls hoping for improvements in weight loss and physical appearance, while boys wanted to build muscle and become good at sport [26].

This finding is based on nine reviews [23–29,31,32] (with 180 included primary qualitative studies). Most of the studies had minor or moderate concerns regarding methodological limitations but the data richness score varied amongst the studies.

**Children and adolescents with obesity can view the idea of diet or physical activity interventions negatively, and question the value of such activities.** Some children and adolescents with obesity viewed the idea of dietary intervention, exercise or physical activity interventions negatively and questioned the value of such activities. However, reasons for questioning the value of these interventions differed between dietary interventions and physical activity interventions. Some children and adolescents with obesity did not engage in physical activity as they gained no pleasure

**Table 6. Overall summary of qualitative findings (diet and physical activity interventions).**

| Supporting studies | Findings |
|---|---|
| **Perceptions of value (for full details see S5 Table)** | |
| **Chen (2024) [31] Jones (2019) [23] Kebbe (2017) [24] Kelleher (2017) [25] Lachal (2013) [26] Lang (2021) [27] Liu (2021) [28] Roberts (2021) [29] Skogen (2022) [19] (n = 9)** | **Children and adolescents can be motivated to engage with obesity health services that focus on diet or physical activity interventions because of the perceived benefits of such activities**<br>— Benefits: weight loss<br>— Benefits: more than just weight loss<br>— Benefits: can differ by gender<br>— Motivations: Observing success in others |
| **Chen (2024) [31] Kebbe (2017) [24] Lachal (2013) [26] Lang (2021) [27] Stankov (2012 [19] Skogen (2022) [19] (n = 6)** | **Children and adolescents can view the idea of diet or physical activity interventions negatively, and question the value of such activities**<br>— Value of intervention questioned |
| **Competing priorities (for full details see S6 Table)** | |
| **Kebbe (2017) [24] Lang (2021) [27] Liu (2021) [28] Roberts (2021) [29] Stankov (2012) [19] (n = 5)** | **Individual children, adolescents and their families have many other commitments, and competing priorities can represent a problem when it comes to engaging in dietary or physical activity interventions**<br>— Opportunities: Absence of structured activities<br>— Opportunities: Insufficient time (for exercise or time to prepare meals, engage and implement intervention, encourage, and discuss healthy diet) |
| **The role of social support (for full details see S7 Table)** | |
| **Chen (2024) [31] Burchett (2018) [21] Jones (2019) [23] Kebbe (2017) [24] Kelleher (2017) [25] Lachal (2013) [26] Lang (2021) [27] Liu (2021) [28] Roberts (2021) [29] Stankov (2012) [19] Skogen (2022) [19] (n = 11)** | **Family involvement and support is important if children and adolescents are to engage with dietary or physical activity interventions**<br>— Concerns over parents' and siblings' levels of engagement with the intervention<br>— Preferences for high levels of engagement with the intervention by parents and siblings<br>— Competing priorities<br>— Parents need to act as good role models by participating in exercise and physical activity<br>— Reliance on parents to provide access to and means of doing adequate physical activity<br>— Reliance on parents for resource to access and to provide means of healthy food including purchasing and preparing healthy meals<br>— Barriers: Challenging family dynamics – conflicting opinions and approaches to lifestyle change |
| **Chen (2024) [31] Jones (2019) [23] Kebbe (2017) [24] Lachal (2013) [26] Lang (2021) [27] Roberts (2021) [29] Stankov (2012) [19] Zarnowiecki (2020) [30] (n = 8)** | **Health worker support is important if children and adolescents are to engage with dietary or physical activity interventions**<br>— Health workers need to provide supportive, structured guidance<br>— Advice and support from health workers is valued<br>— Concerns over health worker levels of engagement with the intervention<br>— Health workers need to provide optimal care, be knowledgeable, share information and be empathetic<br>— Barrier: Negative experience with health workers and lack of support<br>— Concerns over stigmatization |
| **Chen (2024) [31] Burchett (2018) [21] Jones (2019) [23] Kebbe (2017) [24] Kelleher (2017) [25] Lang (2021) [27] Roberts (2021) [29] Stankov (2012) [19] Skogen (2022) [19] (n = 9)** | **Peer involvement and social support is important if children and adolescents are to engage with dietary or physical activity interventions**<br>— Friends and peers need to act as good role models by participating in exercise and physical activity<br>— Friends and peers need to offer support<br>— Friends and peers Involvement and engagement with the intervention makes individual feel a sense of belonging and being accepted<br>— Barrier: Peer pressure, influence of bad behaviour from peers |
| **The environment (for full details see S8 Table)** | |
| **Chen (2024) [31] Jones (2019) [23] Lang (2021) [27] Stankov (2012) [19] (n = 4)** | **Physical resources at school (spaces, equipment and facilities) need to be accessible and appropriate if children and adolescents are to engage with physical activity interventions**<br>— Appropriate changing and exercise facilities need to be accessible at time of intervention and beyond (school) |
| **Chen (2024) [31] Kebbe (2017) [24] Lang (2021) [27] (n = 3)** | **Physical resources at home and in the community (safe spaces to play and exercise) need to be accessible and appropriate if children and adolescents are to engage with physical activity interventions**<br>— Appropriate exercise facilities need to be accessible at time of intervention and beyond (home and community) |

*(Continued)*

**Table 6.** (Continued)

| Supporting studies | Findings |
|---|---|
| **Kebbe (2017) [24] Stankov (2012) [19] (n = 2)** | **If the weather conditions are inappropriate to the type of planned physical activity, then children and adolescents may not be motivated to engage in physical activity**<br>— Opportunities: weather can limit scope for exercise and physical activity |
| **Kelleher (2017) [25] Lang (2021) [27] Liu (2021) [28] Molina (2021) [20] Roberts (2021) [29] Zarnowiecki (2020) [30] (n = 6)** | **Resource, programme and facilities at school, home and in the community need to be accessible for children, adolescents and their families to engage with dietary interventions**<br>— Accessibility to programmes and facilities: Clinic Location<br>— Accessibility to programme and facilities: Timing and frequency of appointments<br>— Accessibility to programme and facilities: Logistic and transport<br>— Availability and access to healthy food: Options at school<br>— Availability and access to healthy food: Social and cultural environment<br>— Resource limitations: lack of internet, difficulty in using app, set-up time |
| **The nature of the diet or physical activity interventions (for full details see S9 Table)** | |
| **Burchett (2018) [21] Jones (2019) [23] Kebbe (2017) [24] Kelleher (2017) [25] Lachal (2013) [26] Lang (2021) [27] Liu (2021) [28] Molina (2021) [20] Roberts (2021) [29] Stankov (2012) [19] Zarnowiecki (2020) [30] (n = 11)** | **The types of diets or physical activities available are important; they should be social, fun, interactive, informal and regular; healthy food should be palatable**<br>Characteristics of intervention:<br>— Informal and fun preferred over highly structured or competitive; preferences for social/ team/ group activities with peers and friends<br>— Preferences for exercise and activity rather than diet as a means of weight management<br>— Preferences for regular or frequent exercise or physical activity<br>— Preferences for group activities and programmes with peers and friends<br>— Preferences for Lifestyle focused intervention, which incorporated nutrition, physical activity and behavioural components.<br>— Intervention should be structured with meal plans and recipe ideas<br>— If intervention includes digital tools, it should be easy to use.<br>— Healthy food should be palatable<br>— Intervention should also include psychological support |
| **Jones (2019) [23] Lachal (2013) [26] Lang (2021) [27] Roberts (2021) [29] Stankov (2012) [19] Zarnowiecki (2020) [30] (n = 6)** | **The types of diets or physical activities available should be appropriate for the child in terms of their age, gender, ethnicity, culture and physical capabilities**<br>Characteristics of intervention<br>— Concerns over required clothing for activities<br>— Perceived as too demanding by professionals and the children and adolescents themselves<br>— Characteristics of intervention: Perceived as too difficult to do at home<br>— Intervention need be tailored and personalised, e.g., portion size, recipes, age appropriate<br>— Dietary strategies need to be suited to the child/ adolescent and to their expectations |
| **Burchett (2018) [21] Jones (2019) [23] Kebbe (2017) [24] Kelleher (2017) [25] Lang (2021) [27] Skogen (2022) [19] Stankov (2012) [19] (n = 7)** | **Children and adolescents feel that it is important to 'fit in' and not to 'stand out'**<br>Characteristics of intervention<br>— Positive experience of being with other children who are the same (normalization)<br>— Social interaction with peers in group session makes the child feel accepted and a sense of belonging<br>— Perceived as stigmatizing, self-conscious<br>— Characteristics of intervention: adopting healthy eating practices with others, such as helping with portion control and sharing food |
| **Burchett (2018) [21] Jones (2019) [23] Haracz (2013) [22] Kebbe (2017) [24] Kelleher (2017) [25] Lang (2021) [27] Roberts (2021) [29] Stankov (2012) [19] Zarnowiecki (2020) [30] (n = 9)** | **The quality and nature of advice on types of diet or physical activity interventions given to children and families is important (what activities might be best, how to do them, goals to aim for)**<br>Characteristics of intervention:<br>— Preferences for the involvement of specialist<br>— Preference for evidence based, trusted and endorsed information<br>— Use of positive language<br>— Needs to give participants new knowledge and skills<br>— Preferences for autonomy in choice of activities rather than the involvement of specialist providers during and after intervention<br>— Practical sessions preferred to theory<br>— Active engagement<br>— Needs to give participants new knowledge and skills (understanding the nutritional content of different foods and drinks as well as giving them a better awareness of what foods should be eaten in moderation)<br>— Prescriptive and regulated diet routine set by a health worker<br>— Use of concise, clear, consistent, direct and practical messages/advice<br>Need for emotional and knowledge support |

*(Continued)*

**Table 6.** (Continued)

| Supporting studies | Findings |
|---|---|
| **Cost (for full details see** S10 Table) | |
| **Kebbe (2017) [24] Lang (2021) [27] Liu (2021) [28] Roberts (2021) [29] Zarnow-iecki (2020) [30] (n = 5)** | **Cost can be a real concern for children, adolescents and their families when seeking to engage with dietary or physical activity interventions**<br>— Concerns over costs and logistics of attendance<br>— Concerns over cost of healthy food (believes healthy food is more expensive)<br>— Concern over cost of tools and apps to engage with intervention |

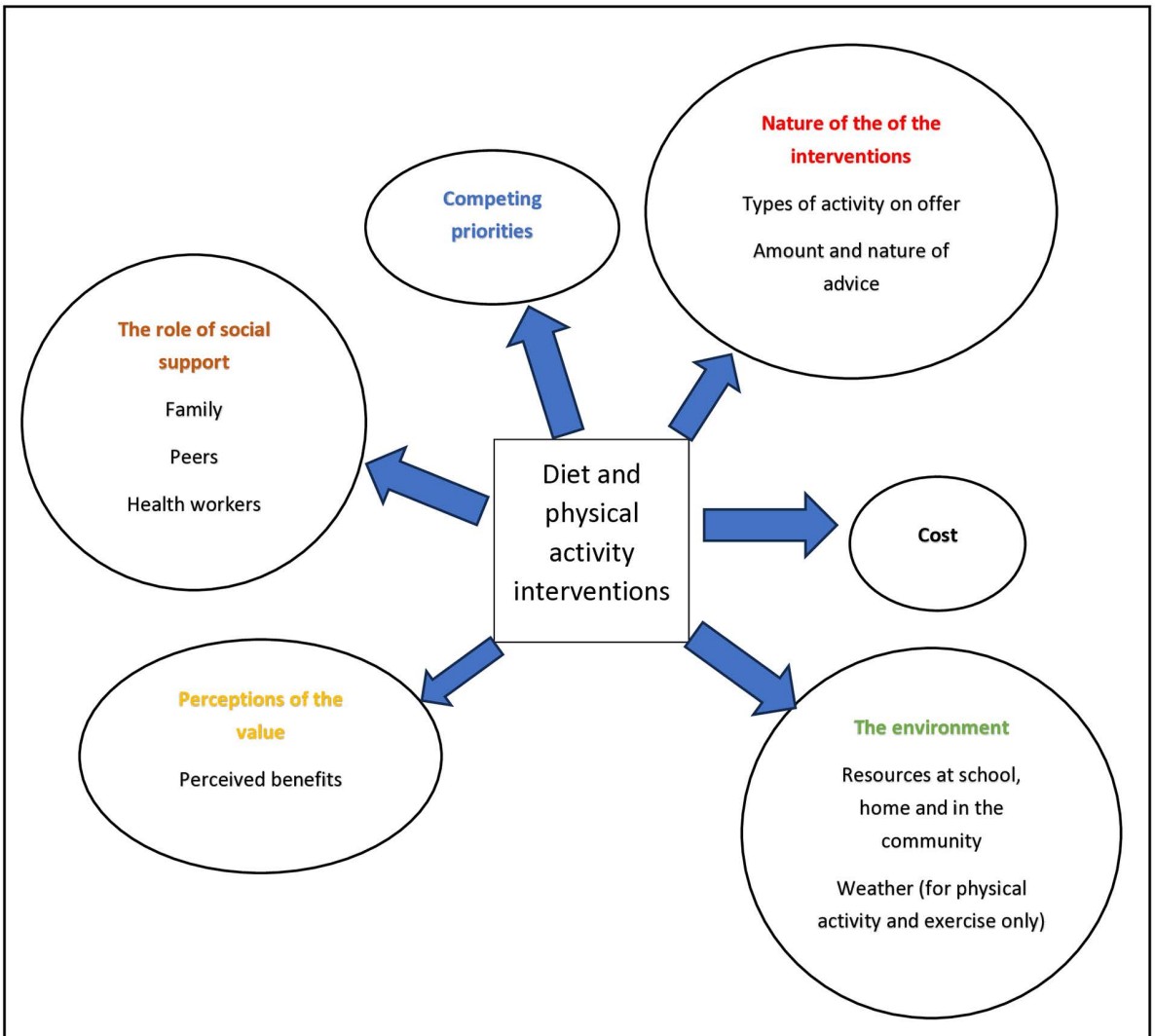

**Fig 2. Factors affecting willingness or ability to engage with diet, physical activity and exercise interventions.**

from exercise, "*I gained weight gradually every year, like five or ten pounds. PE didn't do anything for me*" [24]; while some reported that even the idea of such activities occasioned a negative self-image because they were worried about not being able to do the activities due to their weight or a lack of the necessary skills, "*I think the weight caused [failure in*

*physical education class]. Because I was overweight I didn't want to make an effort. I didn't want to try because I knew I wouldn't be good at it"* [24,26,31,32]. Where the impact of dietary interventions was small and unfavourable with intangible benefit, this reduced adolescents and children's desire to sustain dietary change [26,27].

Some participants reported difficulties in engaging with the intervention during periods of increased stress, the problems of coping with hunger, and a dislike of the lifestyle changes required to maintain weight [27].

This finding is based on six reviews [19,24,26,27,31,32] (with 108 included primary qualitative studies), with minor or moderate concerns regarding methodological limitations, but variable richness data.

### Competing priorities

**Individual children, adolescents and their families have many other commitments; such competing priorities can represent a problem when it comes to thinking about or engaging in diet and physical activity interventions.** Children and adolescents with obesity reported that they had fewer opportunities for structured activity when engaging with exercise and physical activities [24,27], in part because as they became older, they became more autonomous and did less with family and at school; they increasingly just engaged in sedentary activities [27]. They also described having other commitments, such as school work and seeing friends [24] or just lacked the time and energy for such activities, *"...if I exercise, I have to exercise for a long time to burn calories, and I think this will make me very tired"* [27]. Other family members also reported competing priorities of work and family, which reduced the time available for such activities [24]. There was also a reported gender and cultural element: African-American adolescent girls reported that physical activity were simply not seen as a priority: *"[Girls are] not interested in working like real hard doing all that exercise stuff. Some of them like – like girls they don't like to sweat and get their hair messed up...They think like they do the wrong thing, they break their nails, it's a crisis."* [19].

In terms of implementing dietary interventions, parents of children and adolescents with obesity reported that due to their multiple commitments, busy schedule and lack of time, it was difficult to make time for discussions about health and diet with their children [27,29], to implement healthy eating habits [28] or to prepare meals in accordance with the intervention [24,28].

This finding is based on five reviews [19,24,27–29] (with 99 included primary qualitative studies). The methodological quality and the richness of the data varied amongst the studies.

### The role of social support

**Family involvement and support is important if children and adolescents with obesity are to engage with diet or physical activity interventions.** Children and adolescents with obesity reported that they liked parents and family to be involved as much as possible in their dietary management and physical activities; having family actively involved with the intervention (exercise or diet) acted as a motivator to engage and continue engagement with such interventions: *"I don't really go to gyms. At my mom's house, I have a basketball court. When I work out and I do stuff, I usually like to do it with my cousin or by myself... because then I don't have to prove nothing to nobody. I do what I can do".* [24,25,31].

Parents also reported valuing the intervention activities such as physical exercises and/ or diet management [25]. Some children and adolescents – of both genders – reported that they valued the support and involvement of mothers the most. The types of support adolescents preferred from their families differed. Some adolescents with obesity reported that they preferred emotional, motivational or verbal support from family members, including parents and siblings [24,28,32]. Others preferred active participation in the intervention by family members: *"having my mom with me really motivates me"* [24,28]. Children and adolescents reported that the failure of parents and siblings to engage with the intervention, or their apparent lack of interest, understanding or knowledge about the need and value of such activities adversely affected whether they were likely to participate, *"there was a time when my daughter would say, I don't want to go,'cause they're telling me I can't eat this and can't eat that. And I go, No we'll go,'cause they're telling me the same thing. When she saw*

*it was difficult for me too and we started getting into a routine, she started wanting to go"* [31]. Some parents had cultural beliefs suggesting that thinness is a sign of sickness and convinced adolescents that body weight is not an important negative indicator of health [28]. The conflicting opinions and approaches to healthy lifestyle from parents and other family members could also be confusing for children and adolescents, preventing them from engaging successfully with the intervention [29].

Some children and adolescents also reported that parents were important role models: adolescents considered it frustrating when parents failed to model appropriate behaviours, *"How can I tell her "this is what you need to do" if she's not seeing me do it"* [24]. Some adolescents also reported that personal norms, such as family members themselves living with their own obesity, made it more difficult for them to engage with the intervention [19]. Conversely, it was reported that if a family had an active lifestyle, then this made participation with the intervention much easier [25].

Some children and adolescents reported how they depended on their parents either for access to, or to provide the means for participating in, certain exercise and physical activity interventions, such as providing transport to gyms [24,27]. Children and adolescents may also rely on their parents to purchase, prepare and provide intervention-specific food choices [24,28]. Hence, some adolescents described a lack of control and influence over their diet: "*If you're just having something for dinner and it's... healthy or not healthy... it's not like you can change it necessarily because if that's what... is made at home, then that's what you're going to eat"* [24].

By contrast, others reported that being away from home and their parents made it difficult for them to maintain healthy lifestyle.

This finding is based on eleven reviews [19,21,23–29,31,32] (with 206 included primary qualitative studies). Most of the studies had minor or moderate concerns regarding methodological limitations, but concerns regarding richness varied.

**Health worker support is important if children and adolescents are to engage with diet or physical activity interventions.** Children and adolescents with obesity reported that the failure of health workers delivering the intervention, to act as role models [26], or to support children and adolescents with structured advice on diet, nutrition, exercise and physical activities, adversely affected their participation in interventions: "*..the problem was not being taken seriously, and felt rejected when asking for help"* [24,31]. Conversely, sensitive and supportive health workers offered motivation for some children and adolescents: "*I went to you know like my GP a couple of times to try and get advice on …what I should do … [was advised to] be mindful of what the intake was and perhaps to, to exercise regularly you know with, either with friends or you know try and get support you know. So that did help a lot.*". [27].

Adolescents and their parents valued health workers who were experienced, knowledgeable and who specialised in supporting children with obesity [23]. Parents respond best to health workers who are empathetic, direct and clear [26]. Parents viewed good staff-participant relationships as vital for continued attendance voicing their intention to drop out of treatment if staff lack experience, enthusiasm or group management skills. Some adolescents also reported that education and physical exercise professionals, rather than supporting those with obesity, contributed to stigmatization, for example, by punishing the whole class for the slowest person or highlighting their weight as an issue [19].

This finding is based on eight reviews [19,23,24,26,27,29–31] (with 144 included primary qualitative studies). Most of the studies had minor concerns regarding methodological limitations, but variable concerns regarding richness.

**Peer involvement and social support is important if children and adolescents are to engage with diet or physical activity interventions.** Children and adolescents with obesity reported that social support and reaction from friends and peers was highly influential when deciding to engage with, or reject, dietary management or physical activity interventions. Adolescents in one study described being around their peers as a security blanket, allowing them to feel comfortable and confident: "*your friend[s] are there, and they can motivate you to do better, too'*. [23]. Having peer support gave them a sense of belonging by allowing them to talk to others in a similar situation to them, sharing their struggles and issues and this motivated them to take part in the interventions, "*I found them fun because I was surrounded by different people who were in the situation that I was in."* [23,32]. In particular, group sessions were described as having

a positive impact on children's confidence and viewed as fundamental to both initiation and maintenance of health behaviour changes [23,25,29]. Adolescents also reported that peers needed to act as role models [19]. Conversely, the failure of peers to support them in their engagement with healthy diet or physical activity adversely affected the likelihood of their participation with interventions [19,24,31,32] The personal norms of peers could even contribute to the stigmatizing of exercise and physical activities: *"In the gym, they laugh and talk behind my back,"* and, *"It hurts me when they say 'hey there, fat kid.' I try to ignore them, but it does not stop"* [24]. Concerns about stigmatization by peers – either due to the physical activities involved or clothing required, especially for girls – was reported by adolescents as a barrier to engaging with such activities [19,31].

This finding is based on nine reviews [19,21,23–25,27,29,31,32] (with 113 included primary qualitative studies). Most studies had minor or moderate concerns regarding methodological limitations, but concerns regarding richness varied.

### The environment

**Physical resources at school (spaces, equipment and facilities) need to be accessible and appropriate if children and adolescents are to engage with physical activity interventions.** Adolescents with obesity reported that they needed their schools to have accessible and appropriate facilities, both during and after school, if they were to be able to engage with physical activity interventions, and to maintain exercise activities after the programme has ended [19,27,31].

Some adolescent girls with obesity also pointed out that provision of appropriate facilities includes changing room and exercise areas where they do not feel 'on display': *"...the worst bit was getting changed and getting into the uniform for PE,..."* [19,31]. The non-availability of preferred activities was also reported by some adolescents to be an issue: caps on numbers and certain activities only being offered at particular times of the year prevented engagement in preferred exercise interventions [19].

This finding is based on four reviews [19,23,27,31] (with 70 included primary qualitative studies). There were minor to moderate concerns regarding methodological limitations and minor to moderate concerns regarding richness.

**Physical resources at home and in the community (safe spaces to play and exercise) need to be accessible and appropriate if children and adolescents are to engage with physical activity interventions.** Some children and adolescents with obesity reported that they wanted access to physical spaces and facilities both at home [24] and in their community [24,27] to support their engagement with exercise or physical activity interventions, and that such physical spaces should be local and safe, *"I need someone to walk with me. My mom doesn't want me walking around by myself. She says that she doesn't trust the guys in the neighborhood"* [24,31].

This finding is based on three reviews [24,27,31] (with 42 included primary qualitative studies). There were minor to moderate concerns regarding methodological limitations but moderate to serious concerns about richness.

**If the weather conditions are inappropriate for the type of planned physical activity, then children and adolescents may not be motivated to engage in physical activity.** Children and adolescents with obesity report concerns about how excessively hot or cold weather limits their opportunities to engage with exercise or physical activity interventions [19,24].

This finding is based on two reviews [19,24] (with 26 included primary qualitative studies); there were minor concerns regarding methodological limitations, but serious concerns about richness.

**Resource, programme and facilities at school, home and in the community need to be accessible for children, adolescents and their families to engage with dietary interventions.** Adolescents with obesity reported that limited access to intervention-specific food options, as well as facilities that promote healthy eating within school and within the broader social and cultural environment [20,27], prevented them from engaging with dietary management interventions [28]. By contrast, easy access to restaurants and food outlets attracted families to consume 'unhealthy' food [28]. Moreover clinic hours and scheduling conflicts such as school holidays and after-school activities, clinic location and a

lack of transport to programme location and length of visit were reported reasons for families not engaging with the dietary intervention programmes [25,29,30].

This finding is based on six reviews [20,25,27–30] (with 98 included primary qualitative studies). The methodological quality and the richness of the data varied between the reviews.

### The nature of the diet or physical activity intervention

**The types of diets or physical activities available are important; they should be social, fun, informal and regular.** Parents preferred to enrol their children in programmes that focused on lifestyle (i.e., incorporated nutrition, physical activity and behavioural components) over programs with other primary focuses, such as those centered mainly on medication, clinical treatment, or short-term weight loss interventions. They favoured programmes that took a holistic approach to weight management rather than those that focused on weight loss or dieting alone [25]. Adolescents expressed preference for group/team activities and programmes with peers and friends: *"the team games were good. Boost their confidence to join in with their mates. Cause some of these kids are really isolated so they need team sports to encourage them to join in"* [21]. Some adolescents complained the absence of support from peers when exercising [19], although some adolescents also reported that going to the gym by themselves was best [23]. They wanted exercise interventions to be available regularly and frequently, rather than being accessible only intermittently [27] and expressed a greater preference for physical activity interventions over dietary interventions [26].

Children, adolescents and their caregivers also reported that they preferred obesity interventions that were highly practical [21], but also informal and fun: *"It wasn't just like 'you need to do more exercise and you need to eat better' – it actually taught us like how to"* [21,23,24,26]. They did not like exercise and physical activities that were highly structured or competitive [26]. Practical and interactive healthy eating sessions were highly valued such as cooking and using visual approaches, e.g., to illustrate portion size [21]. Parents and adolescents wanted interventions to be structured with meal plans and recipe ideas [24,29] and wanted healthy food that tasted good [20,24].

Overall, obesity management interventions should factor in longer term support that considers the mental health of adolescents and psychological support, as obesity management brings about feelings of failure, guilt and shame [23,29].

This finding is based on eleven reviews [19–21,23–30] (with 204 included primary qualitative studies). Concerns over the methodological limitations of the studies and richness of the data varied between the studies.

**The types of diets or physical activities available should be appropriate for the child in terms of their age, gender, ethnicity, and physical capabilities.** Some children and adolescents with obesity and their parents reported that they wanted weight management programmes to be tailored or personalised [30]. The programme should be appropriate for the child/adolescent's age, ability, culture and ethnicity [23,29]. Adolescents expressed strong feelings about the lack of services aimed at them [23]. They wanted to engage with obesity interventions that were created with their age group in mind, with strategies that matched their needs and expectations, *"I like to ride a bicycle, but I can only do so for a short period of time because I feel very, very tired, and I have no strength."* [29].

Some children and adolescents with obesity reported that they found some exercises too demanding, struggling with them on account of their weight or lack of necessary skills: *"I really try and sometimes give up, but most of the time I watch. It is very hard"* [19,26]. The exercises left some adolescents in pain or discomfort, or too tired to do anything else [19]. As a result, some adolescents reported a history of 'failure', and that they felt frequently judged when in classes: *"I'm not very good at a lot of things we do in PE so I get embarrassed when we do those things"* [19]. These problems adversely affected children and adolescent engagement and participation. Some adolescents also reported that the required exercises or activities were too difficult to do at home [19].

Dietary management interventions should also take into account parental ability to accommodate recommendations; and give detailed plans to follow – parents noted that they wanted specific and relevant content, such as portion sizes for different ages and appropriate recipes that would be suitable for all family members [29].

This finding is based on six reviews [19,23,26,27,29,30] (with 118 included primary qualitative studies). Concerns over the methodological limitations of the studies and richness of the data varied between the studies.

**Children and adolescents report that it is important not to 'stand out'.** Some children and adolescents with obesity and their parents report that engaging in physical activities and dietary programmes can be a positive experience if the group is restricted to children and adolescents with obesity, as this offers a normalizing environment where they feel socially accepted [21,25,27]. Some parents who participated in this study felt it was good to attend and '*speak to other parents who are trying to change things*', while their children '*could make friends with other kids*' who could '*play on the same level*' as their own child [25]. Conversely, standing-out in non-intervention activities was reported by some adolescents to be a highly stigmatizing experience, contributing to negative self-image, especially for girls [19,32].

This finding is based on seven reviews [19,21,23–25,27,32] (with 837 included primary qualitative studies); there were minor-to-moderate concerns regarding methodological limitations but mixed concerns regarding richness.

***The quality and nature of advice on types of diet or physical activity interventions given to children and families is important (what activities might be best, how to do them, goals to aim for).*** Some children and adolescents with obesity, along with their caregivers, reported that interventions should provide them with new skills and knowledge. Some preferred practical, hands-on instruction over theory: *"...you don't want to hear theory when you're a mum. You want to hear real-life experience and what's practical for us"* [21,24,25], while others appreciated gaining evidence-based knowledge [23,25]. Some adolescents report that they liked specialist providers, who could give them structured guidance, including clear advice on goal-setting, structured meal plans, recipes, quantity and quality of foods [23,24,27,29,30]; and they wanted to know exactly what physical activities were best, *"I want to start exercising, but I don't know what exercises to do"* [19,22–24]. In addition, a non-forceful approach from health workers was appreciated; and they placed value on gentle encouragement and receiving support that focused on more than just weight loss, such as self-esteem and well-being [23].

However, some children and adolescents also report that they prefer a degree of autonomy in their choice of activities, want to set their own targets during interventions [23,29], rejecting the need for maintenance of support after the intervention had ended [23].

This finding is based on nine reviews [19,21–25,27,29,30] (with 103 included primary qualitative studies); there were mixed concerns regarding methodological limitations and richness.

## Costs

***Cost can be a real concern for children, adolescents and their families when seeking to engage with dietary interventions or with physical activity interventions.*** Concerns were expressed over costs: not just attending clinics but also accessing necessary resources [29]. Some parents reported that the need to pay for services, or a lack of insurance coverage, made it difficult to engage with the intervention [29]. Other parents were concerned at the cost of using digital tools and other resources [30]. Some parents and adolescents reported that limited finances made it difficult to eat intervention-specific foods or engage in certain exercise and physical activities [24,28].

This finding is based on five reviews [24,27–30] (with 93 included primary qualitative studies); there were mix concerns regarding methodological limitations and richness.

## Discussion

This review of qualitative evidence syntheses (mega-ethnography) explored the perspectives of children and adolescents with obesity, as well as their parents and caregivers, regarding the idea and experience of physical activity interventions and/or dietary interventions to manage obesity. By synthesising the evidence from all relevant reviews and their included primary qualitative studies, the current review creates a broader, deeper overview of the evidence base on this topic to inform evidence-supported guidance. Several factors were identified that affect whether children and adolescents with

obesity are prepared to consider, participate in and continue with physical activity interventions and/or dietary interventions to manage obesity. One of the most consistent findings across the synthesis was the value of social support, particularly from family members and peers, as a facilitator of engagement. Children and adolescents are not always convinced of the value of dietary and or physical interventions. Where they and their family are knowledgeable and/or can see ready benefits from engagement, they may be in favour of engaging with the intervention programmes. However, the interventions must contend with many competing priorities both for the young people and their families. This helps to explain why family involvement and support are crucial if children and adolescents are to engage in dietary and physical interventions.

Involvement and support from health workers and other social support is also key in encouraging engagement with dietary and/or physical interventions and this needs to be enhanced by involvement and social support from peers to help the child or adolescent to fit in and to feel accepted. This is echoed in other studies showing that peer modelling and peer-led activities can enhance engagement and reduce the social isolation often experienced by children with obesity [34]. Designing interventions that include peer mentoring, group-based sessions, or buddy systems may therefore help to foster a sense of inclusion and motivation. Yet, it is not enough simply to navigate young people through the pathway of the intervention; resources and support must also be available post-intervention in the form of long-term and ongoing support. Health workers need to be able to tailor the amount and type of advice to the needs of the young person and their family. This aligns with strong evidence that family-based behavioural treatments are more effective than child-only approaches in managing paediatric obesity [35]. Family involvement promotes consistent messaging, shared goal setting, and environmental support for behaviour change. Social support more broadly has also been shown to moderate intervention effectiveness, particularly in sustaining behaviour change over time [36].

At the same time, financial and environmental barriers, such as the cost of healthy food or lack of safe spaces to exercise, were also widely cited across the synthesis. These barriers highlight the importance of equity-focused interventions, including subsidised programme access, transportation support, and school or community-based facilities. Interventions also need to be adaptable to different cultural and socioeconomic contexts, especially for more diverse and underrepresented populations, if they are to be inclusive and effective [37].

Findings specific to dietary management interventions suggest several key areas of focus. Children and adolescents and their parents or caregivers wanted to be equipped with the necessary skills to prepare tasty, healthy, and low-cost meals. In addition, they valued the intervention to be interactive, social, fun, structured and appropriate to the young person's age, ethnicity, and culture. These practical skills related to food preparation help motivate individuals to make more health-conscious food choices and prioritize nutrition and make healthy eating more accessible and enjoyable.

Findings specific to physical activity interventions highlight the importance of non-competitive activities to ensure inclusivity and reduce pressure, particularly for young people who may feel discouraged by competition. Additionally, the need for adequate resources is emphasized, including safe spaces and equipment for physical activities in both community (where children live) and school settings, with provision for indoor and outdoor activities to account for weather conditions. This reinforces the value of adopting a whole-family or social-ecological approach, where parents are actively engaged in programme activities such as joint meal preparation, family physical activities, and behaviour contracts that support consistency at home.

Reviews included in this synthesis confirm that adolescents highly value active engagement when learning about healthy eating and physical activities; a finding also found in younger children, who prefer practical and interactive experiences, rather than receiving didactic information [21,23]. This has also been echoed in prior research highlighting the importance of experiential learning and skills-based education in promoting health behaviours in children [38]. Other reviews confirm the pivotal role that parents play, not only in the provision of food in the household and participation in physical activity, but also in parent modelling of healthy lifestyles influencing their children's choices [39]. The value of paediatric weight management programmes that seek to educate the whole family is corroborated by many studies [40]. Research findings again suggest that young people and their parents alike prefer services that provide

a patient-and-family centred approach, provided that it is also tailored through the use of individualized information and resources. Given that children who experience obesity in everyday life often face stigmatisation or even bullying [23,26,32,41], it is understandable why group support from peers is perceived as critical in obesity management [21]. However, it is also important to recognise that social comparison, a recognised feature of group approaches, may have unintended effects when a child perceives themselves to be different from the group [41].

Another important finding, that corresponds with findings from health promotion to young people [42], is that improved health is not a principal motivation for adolescents with obesity who attend a weight management intervention [23]. Instead, the motivation to lose weight is more commonly related to a desire to be "normal" and socially desirable rather than preventing health consequences. This is consistent with Self-Determination Theory, which highlights the importance of aligning behaviour change with individuals' intrinsic goals and values [43]. To better support adolescents, interventions could incorporate motivational interviewing (MI) to enhance autonomy and allow young people to identify their own reasons for change. MI has been successfully used in adolescent health interventions to enhance engagement and facilitate behaviour change [44]. Programmes could also allow flexibility and choice in activities, giving adolescents a sense of ownership and control. Motivators for weight loss thus should resonate with the adolescent in the short-term; suggestions include increased clothing choices to attend social gatherings, reduced bullying, fewer unwelcome comments by family members, and, in more affluent contexts, material rewards for weight loss such as jewellery, games, or outings [45].

This review addresses dietary and physical exercise components often included in weight management programmes, however, in isolation these intervention components are unlikely to achieve their full impact. Adolescents often need emotional and psychological support within a weight management programme, alongside nutrition and physical activity [46]. This is confirmed by at least one of the featured reviews included in this synthesis [23]. Support must also extend longer term beyond the engagement phase. Longer-term weight loss interventions may decrease the risk of subsequent development of eating disorders [47] and are affirmed by qualitative data that reveals that young people value longer-term support once they have completed a weight loss intervention [23]. Health services should consider whether more costly longer-term interventions may realise benefits in terms of maintenance of weight loss behaviours that mean that they don't encounter ongoing repeat presentation from young person continuing onwards into adulthood.

## Limitations of review

This review has several important limitations that should be considered when interpreting the findings. The main limitation of this review is the transferability of the findings. Most of the included reviews and studies were from high-income countries in Europe and North America, with limited representation from low- and middle-income countries (LMICs), such as those in Africa, South East Asia, South America or the Western Pacific. This geographic concentration is particularly concerning, given the rising prevalence of childhood obesity in many LMICs, where cultural norms, healthcare infrastructure, and socioeconomic conditions differ substantially. This imbalance highlights a significant gap in the current evidence base and emphasizes the urgent need for more research in LMICs, especially in countries with large populations of children and adolescents affected by obesity, such as India and China. Consequently, the applicability of these findings to non-Western or resource-limited settings remains uncertain and should be explored through regionally tailored research. Second, while many reviews grouped children and adolescents together, limited age-specific analysis restricts understanding of how engagement with obesity management programmes might differ across developmental stages. Only one review focused exclusively on the attitudes or experiences of children who are less than ten years old [21], which raises concerns about whether the perspectives and needs of younger children have been sufficiently captured. This could lead to an underrepresentation of age-specific barriers and facilitators in the findings.

Third, the cultural and contextual variation in acceptability was underexplored. Few studies looked at how factors such as religion, ethnicity, or cultural beliefs shape attitudes towards physical activity or dietary practices. This lack of

granularity limits the capacity of our findings to inform culturally tailored interventions and may obscure important sub-group differences.

Fourth, although eligibility criteria required one qualitative study to be included in source reviews for inclusion in this overview of reviews, only five reviews were identifiable as full qualitative evidence syntheses [19,21,26–28]. These five reviews performed well for both methodological limitations (minor-to-moderate concerns) and richness of data (two rich and three moderate) and made a major contribution to the findings. While many findings from these five reviews were corroborated by reviews with methodological limitations and less richness, this review would have been strengthened by the availability of a greater number of formally conducted qualitative evidence syntheses. Hence, the reliance on a small subset of high-quality qualitative syntheses means the robustness of some conclusions may vary depending on the strength of the contributing evidence.

Additionally, by focusing exclusively on "obesity management" rather than broader public health strategies associated with "obesity prevention", the review may overlook upstream societal and environmental influences on obesity. While some values and preferences relating to obesity management are shared with obesity prevention, only the former have been systematically harvested to populate this review, narrowing the scope and relevance of the findings, particularly for policy-level interventions. Another key limitation is that the focus of this review is on the values and preferences of children, adolescents, parents and caregivers, which means we did not explore the views of other important stakeholders, such as health workers, schoolteachers or other education or social care workers, even though these individuals often play a crucial role in programme design and delivery, and their omission limits the comprehensiveness of the stakeholder perspective. However, their role was referenced by adolescents and caregivers in the included reviews. Finally, the data did not allow examination of specific individual exercise regimes or diet plans, nor differences in types of exercise/activity or diet composition, but focused only on the values and preferences regarding the programmes themselves. As such, it cannot inform detailed decisions about tailoring individual programme elements.

Together, these limitations suggest that while the findings offer valuable insights, they should be interpreted with caution. There is a clear need for more diverse, age-specific, and culturally sensitive qualitative research in this area, particularly in underrepresented regions and populations.

## Implications for practice

Some children and adolescents are highly motivated to participate in diet or physical activity to help with their obesity, but others do not see the value of such activities or are reluctant to be involved.

Practitioners seeking to encourage children and adolescents with obesity to engage and participate in dietary and physical activities can draw upon these findings to design more effective programs. This review suggests that successful interventions will not simply deliver information but will actively address the core needs and preferences of young people and their families. Examples of strategies that could be incorporated in the interventions are listed below:

- For physical activity interventions, prioritize non-competitive activities like dance classes, community-based sports clubs focused on participation rather than winning, or exergames (video games that require physical movement). Ensure safe and accessible spaces, such as community centers with indoor facilities for bad weather, are available.

- To promote social support, design interventions with a whole-family approach, such as family cooking workshops or group outings. Additionally, create peer support components, like mentorship programs where older participants who have successfully managed their weight can guide newer ones.

- For dietary management, go beyond providing lists of "good" foods. Instead, offer hands-on, interactive cooking classes that teach young people and their caregivers practical skills for preparing tasty, healthy, and affordable meals. This can include recipes tailored to their cultural backgrounds and food preferences.

- To address the need for autonomy and intrinsic motivation, integrate motivational interviewing techniques into the delivery of the intervention. This approach empowers young people to identify their own goals and reasons for change, which aligns with their desire for choice and control.

- Recognizing that motivation is often socially driven, interventions can frame weight management in terms of short-term, tangible benefits that resonate with adolescents, such as increased energy for social activities, reduced bullying, or having more clothing options.

By adopting these specific, evidence-supported strategies, practitioners can create interventions that resonate with the lived experiences of children, adolescents, and their families which may likely improve long-term engagement and outcomes.

### Implications for future research

Findings were derived from fourteen reviews drawing upon a global evidence base from across 227 qualitative and mixed method studies, although evidence mostly derived from high income countries. More research from Low- and Middle-Income Countries (LMICs) is needed. Although some included reviews reported population characteristics such as gender, age, ethnicity, and socio-economic status, these factors were largely absent from the evidence and analysis of most existing reviews. Other factors such as disability were not considered. The provision of such research data would facilitate granular analysis and nuanced understanding of factors that might apply to particular groups of young people. In addition, whilst dietary and physical activity interventions have traditionally formed the cornerstone of obesity management, the increasing adoption of pharmacological therapies, particularly GLP-1 receptor agonists warrants scrutiny of the evolving evidence base [48], especially in adolescent populations. It is essential to assess the short-term acceptability of these agents to patients and to examine how this acceptability may influence longer-term outcomes. This includes monitoring for adverse events and evaluating the potential for relapse into unhealthy dietary and physical activity patterns following discontinuation of pharmacotherapy treatment. Furthermore, while pharmacological therapies such as GLP-1 receptor agonists can be effective in promoting weight loss and reducing caloric intake, they may be less effective in supporting other important health goals such as increasing the consumption of nutrient-dense foods, establishing healthy eating habits, or promoting regular physical activity. Delivery of pharmacological therapy in adolescent populations could be complemented with lifestyle behavior change interventions, particularly to achieve sustainable improvements in health outcomes. The findings of this review could be valuable in informing the design of such complementary lifestyle interventions.

### Conclusion

This review identified multifaceted factors that influenced how young people with obesity and their caregivers engaged with physical activity (including exercise) and/or dietary management programmes. Findings suggest that diet and physical activity interventions are more effective when combined with emotional and psychological support. A whole family approach and trusting personalized relationships with healthcare workers appear important for engagement alongside ongoing support from family, peers, and professionals in the form of anticipatory guidance and motivation, which emerged as a common theme. However, further research is needed to confirm these patterns across diverse populations and healthcare systems. Overall, the findings should be interpreted as context-dependent and may not be universally applicable. Since the evidence is predominantly drawn from high-income countries, the generalisability of the conclusions to other settings may be limited.

### Supporting information

**S1 Appendix. Search strategy.**
(DOCX)

**S2 Appendix. Review author reflexivity.**
(DOCX)

**S3 Appendix. PRISMA checklist.**
(DOCX)

**S1 Table. Reviews excluded with reasons.**
(DOCX)

**S2 Table. Data extraction of included reviews (Dietary management).**
(DOCX)

**S3 Table. Data extraction of included reviews (exercise/physical activity management).**
(DOCX)

**S4 Table. Quality assessment of included reviews.**
(DOCX)

**S5 Table. Summary of Qualitative findings: Perceptions of the value (diet and physical activity interventions).**
(DOCX)

**S6 Table. Summary of Qualitative findings: Competing priorities (diet and physical activity interventions).**
(DOCX)

**S7 Table. Summary of Qualitative findings: The role of social support (diet and physical activity interventions).**
(DOCX)

**S8 Table. Summary of Qualitative findings: The environment (exercise and physical activity interventions).**
(DOCX)

**S9 Table. Summary of Qualitative findings: The nature of the diet or physical activity interventions.**
(DOCX)

**S10 Table. Summary of Qualitative findings: Cost of diet and physical activity interventions.**
(DOCX)

## Acknowledgments

When preparing this review, we used EPOC's Protocol and Review Template for Qualitative Evidence Synthesis (Glenton C, Bohren MA, Downe S, Paulsen EJ, Lewin S, on behalf of Effective Practice and Organisation of Care (EPOC). EPOC Qualitative Evidence Synthesis: Protocol and review template. Version 1.1. EPOC Resources for review authors. Oslo: Norwegian Institute of Public Health; 2020. Available at: http://epoc.cochrane.org/epoc-specific-resources-review-authors)

## Author contributions

**Conceptualization:** Christopher Carroll, Andrew Booth.

**Data curation:** Munira Essat, Christopher Carroll, Andrew Booth, Joanna Leaviss, Diana Castelblanco Cuevas.

**Formal analysis:** Munira Essat, Christopher Carroll, Joanna Leaviss, Diana Castelblanco Cuevas.

**Funding acquisition:** Christopher Carroll, Andrew Booth.

**Investigation:** Christopher Carroll, Andrew Booth.

**Methodology:** Christopher Carroll, Andrew Booth.

**Project administration:** Andrew Booth.

**Supervision:** Andrew Booth.

**Validation:** Munira Essat, Christopher Carroll, Andrew Booth, Roos Verstraeten.

**Visualization:** Munira Essat, Christopher Carroll.

**Writing – original draft:** Munira Essat.

**Writing – review & editing:** Munira Essat, Christopher Carroll, Andrew Booth, Joanna Leaviss, Diana Castelblanco Cuevas, Roos Verstraeten.

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
