## [Decision Letter · Decision Letter 0]

5 May 2025

Dear Dr. Essat,

Thank you for submitting your manuscript to PLOS ONE. After careful consideration, we feel that it has merit but does not fully meet PLOS ONE’s publication criteria as it currently stands. Therefore, we invite you to submit a revised version of the manuscript that addresses the points raised during the review process.

We look forward to receiving your revised manuscript.

Kind regards,

AKM Alamgir, PhD

Academic Editor

PLOS ONE

Journal Requirements:

“The Department of Nutrition and Food Safety at the World Health Organization (WHO) commissioned and provided financial support to the University of Sheffield for this work. WHO acknowledges the financial support from the Norwegian Agency for Development Cooperation (NORAD), The Swedish International Development Cooperation Agency (SIDA), The Government of the Grand Duchy of Luxembourg, and the Government of Germany (BMG) to the Department of Nutrition and Food Safety.”

“The Department of Nutrition and Food Safety at the World Health Organization (WHO) commissioned and provided financial support to the University of Sheffield for this work. WHO acknowledges the financial support from the Norwegian Agency for Development Cooperation (NORAD), The Swedish International Development Cooperation Agency (SIDA), The Government of the Grand Duchy of Luxembourg, and the Government of Germany (BMG) to the Department of Nutrition and Food Safety. “

“The Department of Nutrition and Food Safety at the World Health Organization (WHO) commissioned and provided financial support to the University of Sheffield for this work. WHO acknowledges the financial support from the Norwegian Agency for Development Cooperation (NORAD), The Swedish International Development Cooperation Agency (SIDA), The Government of the Grand Duchy of Luxembourg, and the Government of Germany (BMG) to the Department of Nutrition and Food Safety.  “

5. As required by our policy on Data Availability, please ensure your manuscript or supplementary information includes the following:

Reviewers' comments:

Reviewer's Responses to Questions

**Comments to the Author**

1. Is the manuscript technically sound, and do the data support the conclusions?

Reviewer #1: Yes

Reviewer #2: Partly

2. Has the statistical analysis been performed appropriately and rigorously?

Reviewer #1: N/A

Reviewer #2: Yes

3. Have the authors made all data underlying the findings in their manuscript fully available?

Reviewer #1: Yes

Reviewer #2: No

4. Is the manuscript presented in an intelligible fashion and written in standard English?

Reviewer #1: Yes

Reviewer #2: No

Reviewer #1: A very Noble topic for Review

The manuscript acknowledges the predominance of studies from high-income countries. By including more studies from low- and middle-income countries would improve the generalizability of the findings. India and China may have the maximum children and adolescents with obesity. However, the review papers that were included in the analysis have minimal representation from these countries.

Reviewer #2: 1. Scientific Review and Research Ethics

• Strengths:

• The research appears ethically sound, with no overt ethical issues in participant recruitment or data handling.

• The objectives and background are well motivated and framed within the context of public health needs.

• Weaknesses:

• The methodology section lacks sufficient detail to ensure full reproducibility. For example, information about the data collection process, survey tools, or any instruments used is limited.

• Statistical methods are not comprehensively described—clarify which tests were used, why, and include confidence intervals or effect sizes.

2. Data Availability and Transparency

• Weaknesses:

• There is no clear data availability statement, which is a core requirement of PLOS ONE.

• The manuscript does not mention whether datasets are publicly accessible or under restricted access.

3. Manuscript Structure and Clarity

• Strengths:

• The structure follows a standard IMRAD format (Introduction, Methods, Results, and Discussion).

• The abstract provides a useful summary, though it could be tightened for clarity.

• Weaknesses:

• There are grammatical issues and awkward phrasing in several sections (e.g., Introduction and Results).

• Transitions between paragraphs are sometimes abrupt, reducing narrative coherence.

4. Discussion and Conclusion

• Strengths:

• Conclusions are consistent with the reported findings.

• The discussion links findings back to the literature.

• Weaknesses:

• Some conclusions may overstate the findings given the study’s limitations (e.g., generalizability or potential biases).

• Limitations are acknowledged but not deeply discussed.

5. Figures, Tables, and Supplementary Material

• Strengths:

• Tables summarize data well and are mostly easy to interpret.

• Weaknesses:

• Tables lack clear titles and captions in some cases.

• No figures or supplementary files were referenced

**Do you want your identity to be public for this peer review?** For information about this choice, including consent withdrawal, please see our Privacy Policy

Reviewer #1: No

Reviewer #2: **Yes:** Rahma K. Abdelseed

---

## [Author Response · Author response to Decision Letter 1]

16 Jun 2025

Dear Editor and Reviewers,

Thank you for reviewing our manuscript, we welcomed your constructive criticism and have improved the manuscript accordingly.

Below are responses to each comment raised.

Editor’s Comments

Author’s Response: The manuscript has been updated to reflect PLOS ONE’S style requirements

“The Department of Nutrition and Food Safety at the World Health Organization (WHO) commissioned and provided financial support to the University of Sheffield for this work. WHO acknowledges the financial support from the Norwegian Agency for Development Cooperation (NORAD), The Swedish International Development Cooperation Agency (SIDA), The Government of the Grand Duchy of Luxembourg, and the Government of Germany (BMG) to the Department of Nutrition and Food Safety.”

Author’s Response: The Funding statement has been updated as below and included in the cover letter:

The Department of Nutrition and Food Safety at the World Health Organization (WHO) commissioned and provided financial support to the University of Sheffield for this work. WHO acknowledges the financial support from the Norwegian Agency for Development Cooperation (NORAD), The Swedish International Development Cooperation Agency (SIDA), The Government of the Grand Duchy of Luxembourg, and the Government of Germany (BMG) to the Department of Nutrition and Food Safety. There was no additional external funding received for this study.

Author’s Response: We confirm our submission contains all raw data required to replicate the results of our review. The data is available either in the main manuscript or in the supplementary file (see response to point 5). As this is an overview of reviews – a secondary analysis, all data used are secondary data which are publicly available. The search strategy used to identify the studies are provide in the supplementary file. In addition we have added an ethical statement to the methods section of the manuscript (Page 5, line 115).

“The Department of Nutrition and Food Safety at the World Health Organization (WHO) commissioned and provided financial support to the University of Sheffield for this work. WHO acknowledges the financial support from the Norwegian Agency for Development Cooperation (NORAD), The Swedish International Development Cooperation Agency (SIDA), The Government of the Grand Duchy of Luxembourg, and the Government of Germany (BMG) to the Department of Nutrition and Food Safety. “

“The Department of Nutrition and Food Safety at the World Health Organization (WHO) commissioned and provided financial support to the University of Sheffield for this work. WHO acknowledges the financial support from the Norwegian Agency for Development Cooperation (NORAD), The Swedish International Development Cooperation Agency (SIDA), The Government of the Grand Duchy of Luxembourg, and the Government of Germany (BMG) to the Department of Nutrition and Food Safety. “

Author’s Response: Thank you for highlighting this.

The Funding Statement has been removed from the Acknowledgement Section. An updated funding statement has been included in the cover letter. See point 2.

5. As required by our policy on Data Availability, please ensure your manuscript or supplementary information includes the following:

(i) A numbered table of all studies identified in the literature search, including those that were excluded from the analyses.

Author’s Response: Thank you for your comment. In accordance with the journal’s Data Availability policy and the PRISMA 2020 guidelines we included a supplementary table (Supplementary Table S1) listing all studies that were excluded at the full-text screening stage, along with the specific reason(s) for exclusion. This is also consistent with guidance provided in the Cochrane Handbook for Systematic Reviews of Interventions which recommends transparent reporting of study selection decisions. In addition a PRISMA flow diagram has been included as Fig 1, documenting the number of studies identified by the searches and the final number of reviews included.

We confirm that all included studies are published and accessible via the provided references. No unpublished studies were included in the review

(ii) A table of all data extracted from the primary research sources for the systematic review and/or meta-analysis. The table must include the following information for each study:

Author’s Response: A table of all data extracted from the primary research sources has been captured in the main manuscript (Page 12, Table 3 and Page 13, Table 4) and additional data been added to Supplementary file, S2 and S3 Tables.

(iii) If applicable for your analysis, a table showing the completed risk of bias and quality/certainty assessments for each study or outcome. Please ensure this is provided for each domain or parameter assessed. For example, if you used the Cochrane risk-of-bias tool for randomized trials, provide answers to each of the signalling questions for each study. If you used GRADE to assess certainty of evidence, provide judgements about each of the quality of evidence factor. This should be provided for each outcome. An explanation of how missing data were handled.This information can be included in the main text, supplementary information, or relevant data repository. Please note that providing these underlying data is a requirement for publication in this journal, and if these data are not provided your manuscript might be rejected

Author’s Response: A completed quality assessment for each review has been presented in the main manuscript (Page 17, Table 5) and details of signalling questions and assessment are provided in Supplementary file S4 Table.

Reviewer’s comments

Reviewer #1:

6. A very Noble topic for Review

Author’s Response: Thank you

7. The title of the study may be reconsidered. The title may be added Systemic Review, add problems and Remedies in the title. Also the age bracket like 5-18 years rather the generalized “ Children and Adolescents” Current title: Exploring the values and preferences of children and adolescents with obesity and their parents/caregivers concerning diet or physical activity interventions for weight management: Mega-ethnography of qualitative syntheses of systematic reviews

Author’s Response: Thank you for your suggestion. After careful consideration, we believe that the current title reflects the scope and focus of the study as per our PROSPERO protocol. We have intentionally used the broader terms “children and adolescents” to encompass the full age range by the included studies, rather than specifying a narrower age bracket. Additionally, the phrase “mega-ethnography of qualitative syntheses” precisely describes the methodology and nature of the review, which we feel is important to convey clearly in the title. As requested Systematic Review has been removed.

8. The manuscript acknowledges the predominance of studies from high-income countries. By including more studies from low- and middle-income countries would improve the generalizability of the findings. India and China may have the maximum children and adolescents with obesity. However, the review papers that were included in the analysis have minimal representation from these countries.

Author’s Response: Thank you for your comment. We agree that including more studies from low- and middle-income countries would improve the generalisability of the findings. As noted in the Limitations Section of our Discussion, we identified very few eligible reviews including studies from these regions for inclusion in our analysis. This highlights an important gap in the existing literature and underscores the need for further research in low- and middle-income countries, including those with large populations of children and adolescents affected by obesity, such as India and China. The discussion section has been edited to reflect this (see Page 35, line 644)

9. While the manuscript discusses age and gender differences, a more detailed analysis of how these

factors specifically influence engagement with interventions would be valuable.

Author’s Response: Thank you for your valuable comment. We have expanded our Discussion to address this point in more detail “… while many reviews grouped children and adolescents together, limited age-specific analysis restricts understanding of how engagement with obesity management programmes might differ across developmental stages. Only one review focused exclusively on children under ten years old, which raises concerns about whether the perspectives and needs of younger children have been sufficiently captured. This could lead to an underrepresentation of age-specific barriers and facilitators in the findings.” (see Page 35, line 655).

10. The manuscript briefly mentions ethical considerations but could provide a more detailed

discussion on the ethical implications of the research, particularly regarding the involvement of

children and adolescents

Author’s Response: Thank you for your comment. We have reviewed the manuscript and note that ethical considerations were not explicitly discussed in the manuscript. As this study is a systematic review synthesizing existing published studies, it does not involve primary data collection or direct participant involvement. Therefore, traditional ethical considerations such as informed consent etc, are not applicable. However, we acknowledge the importance of ethical issues involved with speaking sensitively to children and adolescents about their weight. The following has been added to the Methods section.

“This review did not require separate ethical approval, as it involved secondary analysis of anonymised data from previously published studies. Accordingly, all included primary studies had obtained appropriate ethical approval.” (Page 5, line 115)

11 The discussion section remains silent on the following

-The Lancet commission on obesity published its recommendations earlier this year (Definition and

diagnostic criteria of clinical obesity - The Lancet Diabetes & Endocrinology). In light of these

findings, the manuscript doesn’t seem contemporary

-In light of widespread, at times off-label use of anti-obesity/chronic weight management therapies

like semaglutide and tirzepatide in adolescents, relevance of non-pharmacological interventions

need to be relooked

Author’s Response: Thank you for your detailed and thoughtful feedback. Although pharmacological therapies fall outside the scope of our current review, we appreciate the importance of the recent Lancet commission’s recommendations on the definition and diagnostic criteria of clinical obesity, as well as the evolving landscape of obesity management, including the increasing use of pharmacological therapies. As such we have added the following to our Discussion Section:

“In addition, whilst dietary and physical activity interventions have traditionally formed the cornerstone of obesity management, the increasing adoption of pharmacological therapies, particularly GLP-1 receptor agonists warrants scrutiny of the evolving evidence base,[40] especially in adolescent populations. It is essential to assess the short-term acceptability of these agents to patients and to examine how this acceptability may influence longer-term outcomes. This includes monitoring for adverse events and evaluating the potential for relapse into unhealthy dietary and physical activity patterns following discontinuation of pharmacotherapy treatment.” (Page 38, line 730)

Reviewer #2:

12. The research appears ethically sound, with no overt ethical issues in participant recruitment or data handling.

Author’s Response: Thank you

13. The objectives and background are well motivated and framed within the context of public health needs.

Author’s Response: Thank you

14. The methodology section lacks sufficient detail to ensure full reproducibility. For example, information about the data collection process, survey tools, or any in

---

## [Decision Letter · Decision Letter 1]

30 Jul 2025

Dear Dr. Essat,

We look forward to receiving your revised manuscript.

Kind regards,

AKM Alamgir, PhD

Academic Editor

PLOS ONE

**Journal Requirements:**

Reviewers' comments:

Reviewer's Responses to Questions

**Comments to the Author**

Reviewer #3: (No Response)

2. Is the manuscript technically sound, and do the data support the conclusions?

Reviewer #3: Partly

3. Has the statistical analysis been performed appropriately and rigorously?

Reviewer #3: N/A

4. Have the authors made all data underlying the findings in their manuscript fully available?

Reviewer #3: Yes

5. Is the manuscript presented in an intelligible fashion and written in standard English?

Reviewer #3: Yes

**Reviewer #3:**  I agree with earlier reviewers that this study addresses an important topic, and the results are valuable for informing the refinement and redesign of lifestyle interventions for children and adolescents with obesity. My suggestions and concerns are mainly related to the study framing and rationale, integration of the study aims and results into the broader literature, and discussion of how the results could be applied.

1. I would recommend that the introduction provide an argument for why it is important to understand the values and preferences of patients, and their respective ‘communities’ (families), in relation to interventions and treatment. This argument can draw on literature from community-engaged research and translational science principles, and the evidence base documenting how program/treatment adoption and efficacy can be improved by ensuring interventions align with patient and community priorities. I would also recommend this be highlighted in the discussion as motivation for why interventionists should use the results from this study.

2. Line 98-99: This sentence is a little confusing: “This paper includes the findings

from two of these reviews, examining the evidence on dietary and physical activity interventions”. What are the “two reviews”? Are they separate reviews on (i) diet interventions for children and adolescents, and (ii) physical activity interventions for children and adolescents? Please clarify.

3. Related to the item above, please clarify in the introduction, methods, and results if the goal of the review was to review interventions that targeted (i) diet only, OR (ii) physical activity only, OR (iii) diet and physical activity jointly (but not other health behaviors).

4. Line 100-101: Please clarify whose ‘values and preferences’ have been excluded (children, parents, both?), as well as clarify what these values and preference are of (e.g., values related to the treatment for obesity in children and adolescents)?

Methods

5. Table 1: There are several aspects of this table that I found unclear or confusing:

-Please clarify what “Setting” refers to. E.g., is the setting in which the intervention is delivered?

-Please clarify why the focal ‘Setting’ for children is “Home” (but not childcare or school), and why the focal ‘Setting’ for adolescents is “School” (but not home). And why is the “Setting” for parents/caregivers only “Clinical settings”?

-Under “Phenomenon of Interest”: use punctuation to make it clear there are two ‘phenomenon’ listed here (Diet intervention; Physical activity intervention)

-I am not clear how to interpret the correspondence of the rows under the “Time/Timing” and “Findings columns: is Time = “Initiation” only applicable to the “Children with Obesity” row; and “Continuation” only applicable to the “Adolescents with obesity row”. I have the same confusion interpreting the rows under “Findings”.

-Line 24, replace “Plus” with “+” so it is clear you are referencing this symbol in “PROGRESS+”

6. Line 137: Revise “for individuals and groups with obesity” to “for individuals with obesity, either delivered in an individual or group setting”. Similarly, could the diet interventions (lines 137-139) be delivered in individual or group formats?

7. Lines 142-143: I do not understand what this sentence means – can you revise for clarity? “Reviews were excluded if the health effects of the interventions could not be attributed distinctly to diet or physical activity or explored diet or physical activity for obesity prevention.” E.g., reviews were only included if the outcomes being measured (e.g., weight loss) were attributed to an intervention component focused on dietary change, or physical activity change, which presumably was a randomized controlled trial testing effects of specific components? It’s unclear how this applies to qualitative research. I also do not understand the point being made in the second part of the sentence, about prevention.

8. Data Extraction (lines 178-189). Could you provide more detail about the method for extracting themes of qualitative analysis results that were summarized in the reviews? E.g., were all themes or topics in the reviews transferred verbatim to a qualitative data management tool, or did any synthesis or transformation of the text about these themes occur at this data extraction stage?

9. Line 195: Is QES an acronym that stands for something? If so, please define it here.

10. Line 238: What does “0/6” years mean, when referring to a child’s age?

11. Lines 241: In the sentence “one review [20] that drew upon evidence exclusively from that region” please state what region this is referring to.

12. Table 3: Under the column “Number of Qualitative studies” please define what the numbers reported represent (i.e., what is the first number, and what is the second bracketed number?).

13. Table 5: Representing the 3 categories as colors (e.g., low = green, orange = medium, red = high) could provide an easier visual interpretation of the patterns of ‘risk to rigor’ across studies and across evaluation features.

14. Table 6: I’d suggest revising the header “The role of support” as “The role of social support”, so it is clear this supporting is coming from interpersonal relationships (and to use this label throughout the manuscript).

15. Figure 2: Why are the arrows between “interventions” and each of the elicited ‘factors that influence engagement in interventions’ bidirectional? The bidirectional arrows imply the factors influence the intervention, and that the intervention influences the factors – is that the intended meaning?

16. Lines 383-396: With regards to the findings on health worker support, can you clarify in this section if the health workers being referred to are predominantly (or only) people who are delivering the intervention, or if the scope of the referent group for health workers is broader and might include health care providers that are not directly providing or delivering the intervention?

17. Line 468: This sentence states “Parents preferred to enrol their children in programmes that focused on lifestyle..”. Please clarify what they preferred this over (i.e., they preferred enrolling their kids in programs focused on lifestyle changes, over what other options/what other programs that have a different focus)?

18. Results: Why are quotes from parents and children so rarely included in the results, given they were included in the data extraction protocol? Their inclusion in lines 516-518 helps to integrate the patient/caregiver voice about their perspectives into the summary of the results. I would recommend including more of these quotes to illustrate key themes.

19. Lines 528-529: The argument in the first few lines of this section is not clear. There are statements that children and caregivers prefer ‘knowledge’, ‘practical instruction’, and ‘evidence-based knowledge’, but they do not prefer ‘theory’. How does ‘evidence based knowledge’ differ to ‘theory’ when provided in the context of the interventions?

20. The discussion rarely integrates the study findings with the broader literature, apart from one paragraph (lines 593-613). I’d suggest that the first few paragraphs not just restate the findings, but interpret them in light of existing evidence and theory, and discuss how they could guide changes to intervention strategies. For example, in relation to the preferences for social support, refer to the extensive literature showing that family-based interventions are more effective than interventions that focus only on the child, and that social support is a known moderator of intervention outcomes; then, provide some examples of how interventions can be design to address this need (e.g., prioritize whole family interventions, include interventions with components to build social support among family and peers). The section “Implications for practice” provides questions that interventionists should ask themselves, but I think it would be useful to additionally provide examples of intervention strategies/protocols/components that actually address specific needs and preferences. E.g. child preferences for autonomy and choice could be addressed by the integration of motivational interviewing in the intervention delivery.

21. Lines 707-709: I think there is an opportunity here to highlight that pharmacological therapies, such as GLPs, can be effective for weight loss and reduced caloric intake, but may not be effective in addressing other health goals: e.g., increasing some healthy dietary behaviors (e.g., consumption of more nutrient dense foods or healthy eating habits, or increased physical activity). Arguably their delivery in adolescent populations should be complemented with lifestyle behavior change interventions, particularly to achieve sustainable improvements in child health outcomes. And so, the results of this study could be very valuable in informing those complimentary lifestyle intervention designs.

what does this mean?). If published, this will include your full peer review and any attached files.

**Do you want your identity to be public for this peer review?** For information about this choice, including consent withdrawal, please see our Privacy Policy

Reviewer #3: **Yes:** Kayla de la Haye

---

## [Author Response · Author response to Decision Letter 2]

23 Oct 2025

Thank you for reviewing our manuscript. We welcomed your constructive criticism and have improved the manuscript accordingly.

Below are responses to each comment raised.

Reviewer’s Comments

Reviewer #3:

I agree with earlier reviewers that this study addresses an important topic, and the results are valuable for informing the refinement and redesign of lifestyle interventions for children and adolescents with obesity. My suggestions and concerns are mainly related to the study framing and rationale, integration of the study aims and results into the broader literature, and discussion of how the results could be applied.

Author Response: Thank you for your helpful suggestions. The manuscript has been updated accordingly.

Reviewer’s Comments

1. I would recommend that the introduction provide an argument for why it is important to understand the values and preferences of patients, and their respective ‘communities’ (families), in relation to interventions and treatment. This argument can draw on literature from community-engaged research and translational science principles, and the evidence base documenting how program/treatment adoption and efficacy can be improved by ensuring interventions align with patient and community priorities. I would also recommend this be highlighted in the discussion as motivation for why interventionists should use the results from this study.

Author Response: Thank you for the helpful suggestion. The Introduction and Discussion section has been updated accordingly.

2. Line 98-99: This sentence is a little confusing: “This paper includes the findings

from two of these reviews, examining the evidence on dietary and physical activity interventions”. What are the “two reviews”? Are they separate reviews on (i) diet interventions for children and adolescents, and (ii) physical activity interventions for children and adolescents? Please clarify.

Author Response: For clarity, this has been amended to - This paper reports the findings of the two systematic reviews that examine the evidence related to (i) dietary interventions for children and adolescents living with obesity, and (ii) physical activity interventions for children and adolescents living with obesity.

3. Related to the item above, please clarify in the introduction, methods, and results if the goal of the review was to review interventions that targeted (i) diet only, OR (ii) physical activity only, OR (iii) diet and physical activity jointly (but not other health behaviors).

Author Response: As suggested, for clarity text has been added to the introduction, methods and results that the goal of the review was to review interventions that targeted (i) diet only and/OR (ii) physical activity only.

4. Line 100-101: Please clarify whose ‘values and preferences’ have been excluded (children, parents, both?), as well as clarify what these values and preference are of (e.g., values related to the treatment for obesity in children and adolescents)?

Author Response: Thank you for your comment. The text has been amended as follows: ... values and preferences of children, adolescents, and/ or their parents/caregivers, that influence engagement and adherence to dietary and physical activity interventions for obesity management in children and adolescents.

Methods

5. Table 1: There are several aspects of this table that I found unclear or confusing:

(i) -Please clarify what “Setting” refers to. E.g., is the setting in which the intervention is delivered?

(ii) -Please clarify why the focal ‘Setting’ for children is “Home” (but not childcare or school), and why the focal ‘Setting’ for adolescents is “School” (but not home). And why is the “Setting” for parents/caregivers only “Clinical settings”?

(iii)-Under “Phenomenon of Interest”: use punctuation to make it clear there are two ‘phenomenon’ listed here (Diet intervention; Physical activity intervention)

(iv) -I am not clear how to interpret the correspondence of the rows under the “Time/Timing” and “Findings columns: is Time = “Initiation” only applicable to the “Children with Obesity” row; and “Continuation” only applicable to the “Adolescents with obesity row”. I have the same confusion interpreting the rows under “Findings”.

(v) -Line 24, replace “Plus” with “+” so it is clear you are referencing this symbol in “PROGRESS+”

Author Response: Thank you for bringing this to our attention. Changes have been made as follows:

(i) Setting (Intervention delivery)

(ii) This is a formatting error and has been resolved

(iii) Punctuation has been added

(iv) This is a formatting error and has been resolved

(v) “Plus” has been replaced with “+”

6. Line 137: Revise “for individuals and groups with obesity” to “for individuals with obesity, either delivered in an individual or group setting”. Similarly, could the diet interventions (lines 137-139) be delivered in individual or group formats?

Author Response: This has been revised as suggested.

7. Lines 142-143: I do not understand what this sentence means – can you revise for clarity? “Reviews were excluded if the health effects of the interventions could not be attributed distinctly to diet or physical activity or explored diet or physical activity for obesity prevention.” E.g., reviews were only included if the outcomes being measured (e.g., weight loss) were attributed to an intervention component focused on dietary change, or physical activity change, which presumably was a randomized controlled trial testing effects of specific components? It’s unclear how this applies to qualitative research. I also do not understand the point being made in the second part of the sentence, about prevention.

Author Response: Thank you. We have amended the paragraph as follows:

The focus of this review was on qualitative data describing the values and preferences of children and adolescents, and of their parents/caregivers, regarding these types of interventions generally. Reviews focusing only on a specific type of exercise or physical activity, or composition of diet were excluded. Furthermore, reviews exploring diet or physical activity for prevention of obesity (i.e. in population who were not obese) were excluded.

8. Data Extraction (lines 178-189). Could you provide more detail about the method for extracting themes of qualitative analysis results that were summarized in the reviews? E.g., were all themes or topics in the reviews transferred verbatim to a qualitative data management tool, or did any synthesis or transformation of the text about these themes occur at this data extraction stage?

Author Response: Thank you for your suggestion. We have now included additional text to clarify the process. Furthermore, this process is, in part, depicted in the supplementary tables, where the third and related fourth-order constructs, and any supporting first order constructs (illustrative quotations) are provided.

9. Line 195: Is QES an acronym that stands for something? If so, please define it here.

Author Response: Thanks for highlighting this. QES – has been written in full: Qualitative Evidence Syntheses

10. Line 238: What does “0/6” years mean, when referring to a child’s age?

Author Response: Thank you. This was a typo and has been amended

11. Lines 241: In the sentence “one review [20] that drew upon evidence exclusively from that region” please state what region this is referring to.

Author Response: Thank you, this has been amended as follows:

one review [20] that drew upon evidence exclusively from Western Pacific region.

12. Table 3: Under the column “Number of Qualitative studies” please define what the numbers reported represent (i.e., what is the first number, and what is the second bracketed number?).

Author Response: Thank you - This has been added: (Number of total studies in the review)

13. Table 5: Representing the 3 categories as colors (e.g., low = green, orange = medium, red = high) could provide an easier visual interpretation of the patterns of ‘risk to rigor’ across studies and across evaluation features.

Author Response: Thank you for the helpful suggestion. This has now been done

14. Table 6: I’d suggest revising the header “The role of support” as “The role of social support”, so it is clear this supporting is coming from interpersonal relationships (and to use this label throughout the manuscript).

Author Response: This has been revised throughout the manuscript

15. Figure 2: Why are the arrows between “interventions” and each of the elicited ‘factors that influence engagement in interventions’ bidirectional? The bidirectional arrows imply the factors influence the intervention, and that the intervention influences the factors – is that the intended meaning?

Author Response: Thank you for highlighting this.

The original intention of the bidirectional arrows was to indicate potential dynamic interactions between interventions and influencing factors; however, we acknowledge that this may have introduced ambiguity regarding causal direction. To improve clarity and align with the primary focus of the figure i.e., identifying the factors that influence engagement in interventions. we have amended the figure so that the arrows point from "interventions" to the influencing factors only.

16. Lines 383-396: With regards to the findings on health worker support, can you clarify in this section if the health workers being referred to are predominantly (or only) people who are delivering the intervention, or if the scope of the referent group for health workers is broader and might include health care providers that are not directly providing or delivering the intervention?

Author Response: Thankyou – The text has been amended for clarity as follows:

The findings are referring to health workers delivering the session – This has been amended in text for clarity.

17. Line 468: This sentence states “Parents preferred to enrol their children in programmes that focused on lifestyle..”. Please clarify what they preferred this over (i.e., they preferred enrolling their kids in programs focused on lifestyle changes, over what other options/what other programs that have a different focus)?

Author Response: Thank you. For clarity the following text has been added:

“Parents preferred to enrol their children in programmes that focused on lifestyle (i.e., incorporated nutrition, physical activity and behavioural components) over programs with other primary focuses, such as those centered mainly on medication, clinical treatment, or short-term weight loss interventions.”

18. Results: Why are quotes from parents and children so rarely included in the results, given they were included in the data extraction protocol? Their inclusion in lines 516-518 helps to integrate the patient/caregiver voice about their perspectives into the summary of the results. I would recommend including more of these quotes to illustrate key themes.

Author Response: Quotes from parents and children have been added to the result section.

19. Lines 528-529: The argument in the first few lines of this section is not clear. There are statements that children and caregivers prefer ‘knowledge’, ‘practical instruction’, and ‘evidence-based knowledge’, but they do not prefer ‘theory’. How does ‘evidence based knowledge’ differ to ‘theory’ when provided in the context of the interventions?

Author Response: For clarity the sentence has been amended as follows:

Some children and adolescents with obesity, along with their caregivers, reported that interventions should provide them with new skills and knowledge. Some preferred practical, hands-on instruction over theory “...you don’t want to hear theory when you’re a mum. You want to hear real-life experience and what’s practical for us” , while others appreciated gaining evidence-based knowledge

20. The discussion rarely integrates the study findings with the broader literature, apart from one paragraph (lines 593-613). I’d suggest that the first few paragraphs not just restate the findings, but interpret them in light of existing evidence and theory, and discuss how they could guide changes to intervention strategies. For example, in relation to the preferences for social support, refer to the extensive literature showing that family-based interventions are more effective than interventions that focus only on the child, and that social support is a known moderator of intervention outcomes; then, provide some examples of how interventions can be design to address this need (e.g., prioritize whole family interventions, include interventions with components to build social support among family and peers). The section “Implications for practice” provides questions that interventionists should ask themselves, but I think it would be useful to additionally provide examples of intervention strategies/protocols/components that actually address specific needs and preferences. E.g. child preferences for autonomy and choice could be addressed by the integration of motivational interviewing in the intervention delivery.

Author Response: We thank the reviewer for this helpful suggestion. We have revised the Discussion to integrate the findings with existing literature and relevant theoretical frameworks (e.g., family-based intervention effectiveness, social support as a moderator, Self-Determination Theory).

We have also added examples of intervention strategies that align with the preferences and needs identified in the review (e.g., motivational interviewing, peer mentoring, family cooking sessions).

Additionally, the “Implications for Practice” section has been updated to include examples of programme components that address specific barriers and motivators.

21. Lines 707-709: I think there is an opportunity here to highlight that pharmacological therapies, such as GLPs, can be effective for weight loss and reduced caloric intake, but may not be effective in addressing other health goals: e.g., increasing some healthy dietary behaviors (e.g., consumption of more nutrient dense foods or healthy eating habits, or increased physical activity). Arguably their delivery in adolescent populations should be complemented with lifestyle behavior change interventions, particularly to achieve sustainable improvements in child health outcomes. And so, the results of this study could be very valuable in informing those complimentary lifestyle intervention designs.

Author Response: Thank you for the suggestion. The following has been added:

Furthermore, while pharmacological therapies such as GLP-1 receptor agonists can be effective in promoting weight loss and reducing caloric intake, they may be less effective in supporting other important health goals such as increasing the consumption of nutrient-dense foods, establishing healthy eating habits, or promoting regular physical activity. Delivery of pharmacological therapy in adolescent populations could be complemented with lifestyle behavior change interventions, particularly to achieve sustainable improvements in health outcomes. The findings of this review could be valuable in informing the design of such complementary lifestyle interventions.

---

## [Decision Letter · Decision Letter 2]

9 Nov 2025

Dear Dr. Essat,

Thank you for submitting your manuscript to PLOS ONE. After careful consideration, we feel that it has merit but does not fully meet PLOS ONE’s publication criteria as it currently stands. Therefore, we invite you to submit a revised version of the manuscript that addresses the points raised during the review process.

We look forward to receiving your revised manuscript.

Kind regards,

AKM Alamgir, PhD

Academic Editor

PLOS ONE

Journal Requirements:

Reviewers' comments:

Reviewer's Responses to Questions

**Comments to the Author**

Reviewer #4: All comments have been addressed

2. Is the manuscript technically sound, and do the data support the conclusions?

Reviewer #4: Yes

3. Has the statistical analysis been performed appropriately and rigorously?

Reviewer #4: Yes

4. Have the authors made all data underlying the findings in their manuscript fully available?

Reviewer #4: Yes

5. Is the manuscript presented in an intelligible fashion and written in standard English?

Reviewer #4: Yes

Reviewer #4: The research article is a good synthesis of existing researches and reviews on managing obesity in children and adolescents, the preferences of these young people and their families regarding interventions focused on managing diet and physical activities. This review can help in shaping interventions according to the preferences of the people who are affected and are to be catered for. This review is supported by evidences are from lived experiences of people who have been directly affected, it has to be given a high standing and urgency in getting published or disseminated as it can help many.

The review is very comprehensive and informative with regard to the topic at hand. The data extracted from other reviews have been presented in a very clear and concise manner yet maintaining the integrity and completeness. The language used to report is very simple and clear making it accessible to all type of readers.

I congratulate the authors in completing a very important study which will add value to existing knowledge and help young people living with obesity.

I do have few suggestions which are minor:

General observation: The search methods and eligibility criteria described is very nice and comprehensive. However, I could not find any guiding research questions in the introduction, objectives or purpose to help in finding relevant reviews and articles and guide the whole process of the research.

76 Is it possible to specify the geographical location? Whether it is globally or in America or Europe? Similarly, regarding the geographic scope of this review, I understand it includes reviews and studies across continents. Is this review aiming to inform WHO regarding the preferences and values of young people and their caregiver’s globally? Although it does mention it cannot be generalized for every setting as the studies included are mostly from high income countries in the limitation. Hence, is it possible to give a geographic scope to the review?

142 Comma (,) after the word inclusion

143 Instead of “Mixed methods reviews were included”, which although correct is a little awkward, is it possible to change it to” Reviews that used mixed methods approach were included”.

316 The sentence structure is a little awkward to read. Maybe rephrase “such as to prevent health sequelae “to “such as preventing health sequelae”.

rephrase “to be” to “being”

rephrase “to gain” to “gaining”

35 Add the word “their” in front of the word “families”

Add the word “other” between the words “many commitments”

354 Change the word “few” to a comparison word such as “fewer” or rephrase the sentence to a comparison sentence.

382 The starting sentence is a little unclear. What do you mean by “such activities”? “The intervention activities such as physical exercises? Or diet management?

393 The information” This could create a sense of self-blaming.” Needs to be continued or explained as the following sentence does not explain it or connect with it.

411 Delete the word “the” in front of the word “home”

512 “On offer” sounds more casual. Can we change it to “available”

521 Is it possible to change “rued” to “complained” or another similar word. “Rued” sounds a little dramatic.

540 “On offer” if we can change it to another appropriate word

578 The title “The amount and nature….” This section does not specify anything about the amount of advice given. In my opinion it is more about the quality or nature of advice on types of intervention.

581 Delete the letter “e” between the words “provide” and “them,

598 Maintain the consistency and add “adolescents “to the topic as well?

what does this mean?). If published, this will include your full peer review and any attached files.

**Do you want your identity to be public for this peer review?** For information about this choice, including consent withdrawal, please see our Privacy Policy

Reviewer #4: **Yes:** Pema Yangzom

---

## [Author Response · Author response to Decision Letter 3]

24 Nov 2025

Reviewer #4:

Reviewer's comment: General observation: The search methods and eligibility criteria described is very nice and comprehensive. However, I could not find any guiding research questions in the introduction, objectives or purpose to help in finding relevant reviews and articles and guide the whole process of the research.

Author's response: Thank you for this insightful observation. We agree that while our aim was stated, explicit research questions were missing, which are necessary to ground the search strategy and synthesis.

We have revised the final paragraph of the Introduction to explicitly list the two research questions that guided this review. These questions focus on identifying specific values and preferences, and understanding how they impact adherence and engagement.

Reviewer's comment: 76 Is it possible to specify the geographical location? Whether it is globally or in America or Europe? Similarly, regarding the geographic scope of this review, I understand it includes reviews and studies across continents. Is this review aiming to inform WHO regarding the preferences and values of young people and their caregiver’s globally? Although it does mention it cannot be generalized for every setting as the studies included are mostly from high income countries in the limitation. Hence, is it possible to give a geographic scope to the review?

Author's Response: Thank you for the suggestion. We have made it clearer the specific geographical location and geographic scope of this review in both the introduction and method section.

Reviewer's comment: 142 Comma (,) after the word inclusion

Author's response: Comma added as requested

Reviewer's comment: 143 Instead of “Mixed methods reviews were included”, which although correct is a little awkward, is it possible to change it to” Reviews that used mixed methods approach were included”.

Author's response: Rephrased as suggested

Reviewer's comment: 316 The sentence structure is a little awkward to read. Maybe rephrase “such as to prevent health sequelae “to “such as preventing health sequelae”. rephrase “to be” to “being”

rephrase “to gain” to “gaining”

Author's response: Rephrased as suggested

Reviewer's comment: 351 Add the word “their” in front of the word “families”

Add the word “other” between the words “many commitments”

Author's response: This has been done as suggested

Reviewer's comment: 354 Change the word “few” to a comparison word such as “fewer” or rephrase the sentence to a comparison sentence.

Author's response: Rephrased to “fewer”

Reviewer's comment: 382 The starting sentence is a little unclear. What do you mean by “such activities”? “The intervention activities such as physical exercises? Or diet management?

Author's response: The sentence has been re-worded as suggested

Reviewer's comment: 393 The information” This could create a sense of self-blaming.” Needs to be continued or explained as the following sentence does not explain it or connect with it.

Author's response: Thank you for highlighting this – it is an error and has been removed.

Reviewer's comment: 411 Delete the word “the” in front of the word “home”

Author's response: “the” has been deleted as suggested

Reviewer's comment: 512 “On offer” sounds more casual. Can we change it to “available”

Author's response: The change has been done as suggested

Reviewer's comment: 521 Is it possible to change “rued” to “complained” or another similar word. “Rued” sounds a little dramatic.

Author's response: “Rued” has been changed to “complained” as suggested

Reviewer's comment: 540 “On offer” if we can change it to another appropriate word

Author's response: The word “on offer” has been changed to “available”

Reviewer's comment: 578 The title “The amount and nature….” This section does not specify anything about the amount of advice given. In my opinion it is more about the quality or nature of advice on types of intervention.

Author's response: This has been re-worded to “The quality and nature of advice on types of diet or physical activity interventions given to children and families is important”

Reviewer's comment: 581 Delete the letter “e” between the words “provide” and “them,

Author's response: Letter “e” has been deleted

Reviewer's comment: 598 Maintain the consistency and add “adolescents “to the topic as well?

Author's response: As suggested “adolescents” has been added to the topic

---

## [Decision Letter · Decision Letter 3]

29 Dec 2025

Exploring the values and preferences of children and adolescents with obesity and their parents/caregivers concerning diet or physical activity interventions for weight management: Mega-ethnography of qualitative syntheses

PONE-D-24-58575R3

Dear Dr. Essat,

We’re pleased to inform you that your manuscript has been judged scientifically suitable for publication and will be formally accepted for publication once it meets all outstanding technical requirements.

Kind regards,

AKM Alamgir, PhD

Academic Editor

PLOS One

Additional Editor Comments (optional):

Reviewers' comments:

Reviewer's Responses to Questions

**Comments to the Author**

Reviewer #1: All comments have been addressed

2. Is the manuscript technically sound, and do the data support the conclusions?

Reviewer #1: Yes

3. Has the statistical analysis been performed appropriately and rigorously?

Reviewer #1: Yes

4. Have the authors made all data underlying the findings in their manuscript fully available?

Reviewer #1: Yes

5. Is the manuscript presented in an intelligible fashion and written in standard English?

Reviewer #1: Yes

Reviewer #1: The submitted manuscript has been reviewed and found it satisfactory.

The author has incorporated all the suggestions

what does this mean?). If published, this will include your full peer review and any attached files.

**Do you want your identity to be public for this peer review?** For information about this choice, including consent withdrawal, please see our Privacy Policy

Reviewer #1: **Yes:** Rakesh kumar gupta

---

## [Editor Report · Acceptance letter]

PONE-D-24-58575R3

PLOS One

Dear Dr. Essat,

I'm pleased to inform you that your manuscript has been deemed suitable for publication in PLOS One. Congratulations! Your manuscript is now being handed over to our production team.

Kind regards,

on behalf of

Dr AKM Alamgir

Academic Editor

PLOS One